# On Path to Multimodal Historical Reasoning: HistBench and HistAgent

**Jiahao Qiu** [* 1]   **Fulian Xiao** [* 2]   **Yimin Wang** [* 3 4]   **Yuchen Mao** [* 4]   **Yijia Chen** [* 5]   **Xinzhe Juan** [3 4]   **Siran Wang** [2]
**Xuan Qi** [6]   **Tongcheng Zhang** [4]   **Zixin Yao** [7]   **Jiacheng Guo** [1]   **Yifu Lu** [1]   **Charles Argon** [8]   **Jundi Cui** [2]
**Daixin Chen** [5]   **Junran Zhou** [2]   **Shuyao Zhou** [1]   **Zhanpeng Zhou** [4]   **Ling Yang** [1]   **Shilong Liu** [1]   **Hongru Wang** [9]
**Kaixuan Huang** [1]   **Xun Jiang** [10 11]   **Xi Gao** [† 2]   **Mengdi Wang** [† 1]

## Dataset Contributors

Yuming Cao, Yue Chen, Yunfei Chen, Zhengyi Chen, Ruowei Dai, Mengqiu Deng, Jiye Fu, Yunting Gu, Zijie Guan, Zirui Huang, Xiaoyan Ji, Yumeng Jiang, Delong Kong, Haolong Li, Jiaqi Li, Ruipeng Li, Tianze Li, Zhuoran Li, Haixia Lian, Mengyue Lin, Xudong Liu, Jiayi Lu, Jinghan Lu, Wanyu Luo, Ziyue Luo, Zihao Pu, Zhi Qiao, Ruihuan Ren, Liang Wan, Ruixiang Wang, Tianhui Wang, Yang Wang, Zeyu Wang, Zihua Wang, Yujia Wu, Zhaoyi Wu, Hao Xin, Weiao Xing, Ruojun Xiong, Weijie Xu, Yao Shu, Yao Xiao, Xiaorui Yang, Yuchen Yang, Nan Yi, Jiadong Yu, Yangyuxuan Yu, Huiting Zeng, Danni Zhang, Yunjie Zhang, Zhaoyu Zhang, Zhiheng Zhang, Xiaofeng Zheng, Peirong Zhou, Linyan Zhong, Xiaoyin Zong, Ying Zhao, Zhenxin Chen, Lin Ding, Xiaoyu Gao, Bingbing Gong, Yichao Li, Yang Liao, Guang Ma, Tianyuan Ma, Xinrui Sun, Tianyi Wang, Han Xia, Ruobing Xian, Gen Ye, Tengfei Yu, Wentao Zhang, Yuxi Wang

## Abstract

Recent advances in large language models (LLMs) have led to remarkable progress across various domains, yet their capabilities in the humanities, particularly history, remain underexplored. Historical reasoning poses unique challenges for LLMs, involving multimodal source interpretation, temporal inference, and cross-linguistic analysis. Existing general-purpose agents perform well on many current benchmarks but lack the domain expertise needed to address complex historical questions. To address this gap, we introduce HistBench, a new benchmark of 414 high-quality and carefully-reviewed questions stratified by difficulty and designed to evaluate LLM's capacity for historical reasoning. The tasks span a wide range of historical problems—from factual retrieval based on primary sources to interpretive analysis of manuscripts and images, to interdisciplinary challenges involving archaeology, linguistics, or cultural history. Furthermore, the benchmark dataset spans 29 ancient and modern languages and covers a wide range of historical periods and world regions. Finding the poor performance of LLMs and other agents on HistBench, we further present HistAgent, a history-specific agent equipped with carefully designed tools for OCR, translation, archival search, and image understanding in history. On HistBench, HistAgent based on GPT-4o achieves an accuracy of 28.50% pass@1 and 36.47% pass@2, significantly outperforming LLMs with online search and generalist agents, including GPT-4o (18.60%), DeepSeek-R1 (14.98%), Grok 3 (17.63%) and Open Deep Research by smolagents (19.57% pass@1 and 25.12% pass@2). These results highlight the limitations of existing LLMs and generalist agents and demonstrate the advantages of HistAgent for historical reasoning. Notably, HistAgent also achieves 60.00% pass@1 accuracy on the GAIA benchmark, showing that domain-specific customization doesn't hinder HistAgent's competitive performance on real-world general tasks. Code is available at https://github.com/CharlesQ9/HistAgent.

[*]Equal contribution [1]AI Lab, Princeton University [2]Department of History, Fudan University [3]University of Michigan [4]Shanghai Jiao Tong University [5]School of Philosophy, Fudan University [6]IIIS, Tsinghua University [7]Department of Philosophy, Columbia University [8]Department of History, Princeton University [9]University of Edinburgh [10]Tianqiao and Chrissy Chen Institute [11]Theta Health Inc.. Correspondence to: Xi Gao <gaoxi@fudan.edu.cn>, Mengdi Wang <mengdiw@princeton.edu>.

*Proceedings of the $43^{rd}$ International Conference on Machine Learning*, Seoul, South Korea. PMLR 306, 2026. Copyright 2026 by the author(s).

## 1. Introduction

Large language models (LLMs) have enabled agents that perform well on diverse complex tasks (Wang et al., 2026), with notable advances in general-purpose and scientific do-

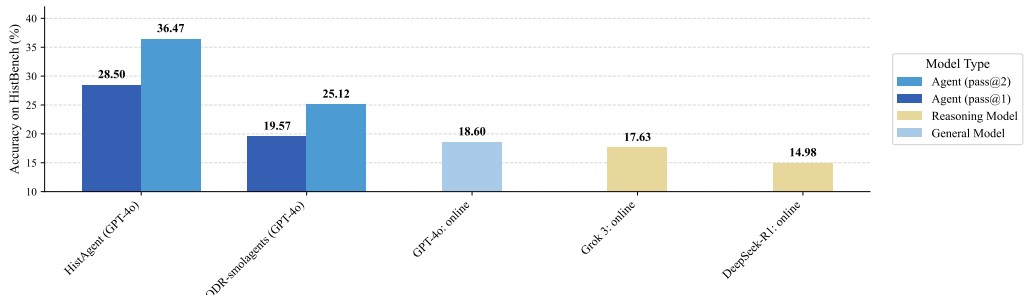

*Figure 1.* Performance of LLMs and Agents on HistBench.

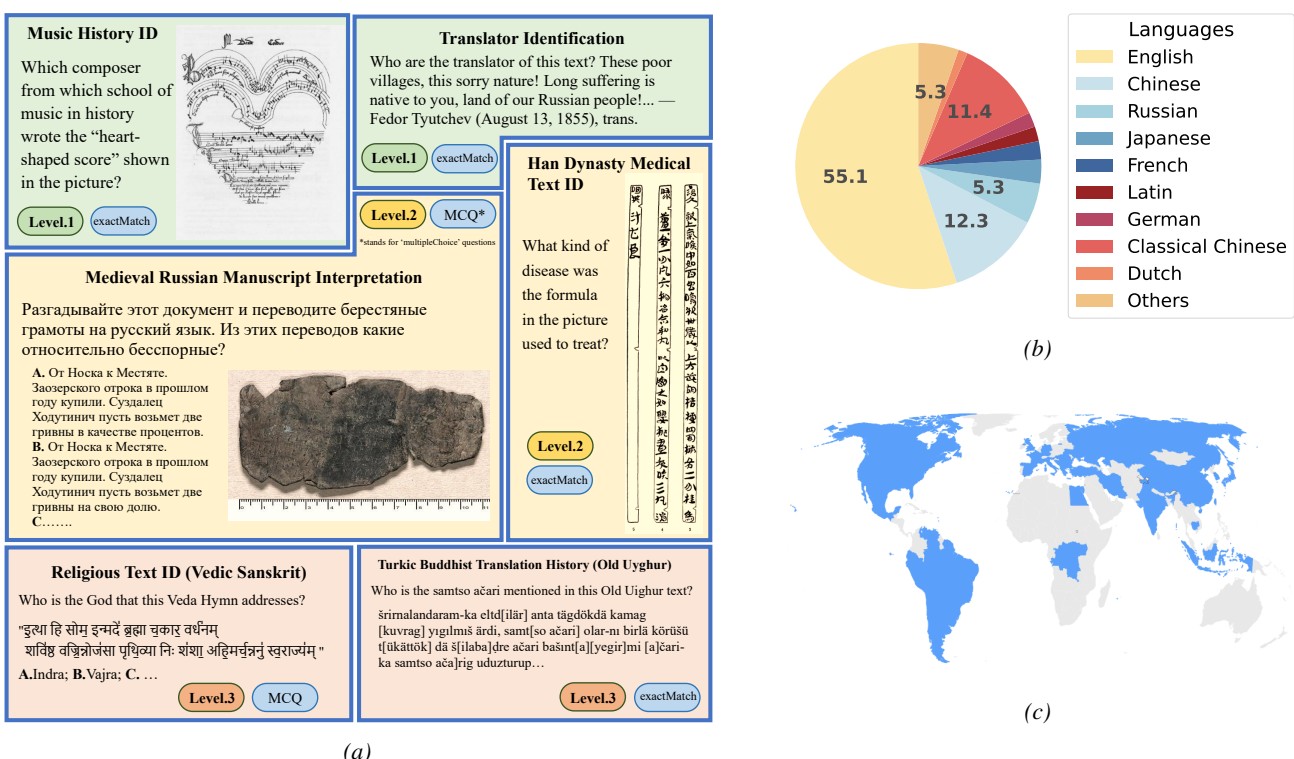

*Figure 2.* **Dataset composition and diversity.** (a) Six representative questions covering both exact-match and multiple-choice formats across Levels 1–3, deliberately chosen to show major source types and scripts: music score, medieval Russian manuscript, Han-dynasty medical text, Vedic Sanskrit hymn, and Old Uyghur translation. These illustrate OCR, translation, and multimodal reasoning challenges central to historical tasks; (b) Language distribution showing 29 modern and ancient languages, capturing both widely used academic and historically significant low-resource languages; (c) Geographic coverage illustrating global reach, with questions from all inhabited continents and deliberate inclusion of underrepresented regions to support diverse historical research.

mains such as Deep Research (OpenAI, 2025) and agents for chemistry and biology (Huang et al., 2024; Bran et al., 2023). Yet the humanities remain largely unexplored, leaving important reasoning challenges unaddressed.

Among the humanities, history holds a uniquely central role. It probes human identity, continuity, and change, while exemplifying the interpretive complexity of humanistic scholarship. Historical reasoning involves incomplete and heterogeneous evidence — from manuscripts and inscriptions to maps and visual records — and requires cross-linguistic, cross-modal, and cross-cultural analysis. At the same time, history produces large, well-annotated textual and visual corpora, making it more computationally tractable than many other humanities fields and an effective testbed for real-world reasoning.

Despite the proliferation of benchmarks, none rigorously target historical reasoning. General-purpose benchmarks such as GAIA (Mialon et al., 2023) include only broad real-world tasks; Humanity's Last Exam (Phan et al., 2025) contains few history questions(only 56 history-related problems); and

*Table 1.* Comparison of historical question-answering datasets. HistBench is the first benchmark specifically curated for historical reasoning in the humanities, while other rows are history-related subsets of general-purpose datasets, highlighting the lack of dedicated history benchmarks. *Hum.* denotes Humanities.

| Dataset | Language | Multimodal | Annotation Source | #Questions | Avg. Accuracy |
|---|---|---|---|---|---|
| MMLU-history subset (Hendrycks et al., 2020) | English | ✗ | Human | 606 | GPT-3 X-Large (Hum.): 40.8% |
| MMLU-Pro-history subset (Wang et al., 2024) | English | ✗ | Machine+Human | 381 | GPT-4: 70.1% |
| C-Eval-history subset (Huang et al., 2023) | Chinese | ✗ | Human | 601 | GPT-4 (Hum.): 62.5% |
| CMMLU-history subset (Li et al., 2023) | Chinese | ✗ | Human | 484 | GPT-4 (Hum.): 72.1% |
| HLE-history subset (Phan et al., 2025) | English | ✓ | Human | 56 | GPT-4o: 2.7% |
| HiST-LLM (Hauser et al., 2024) | English | ✗ | Machine+Human | 36,577 | GPT-4o: 43.7% |
| **HistBench (Ours)** | **29 languages** | **✓** | **Human** | **414** | **GPT-4o (online): 18.6%** |

while domain-specific efforts like PHYBench (Qiu et al., 2025) show the value of tailored evaluation in other areas, no equivalent exists for history. This gap prevents systematic assessment of models on the distinctive methodological demands of historical research.

To address this gap, we introduce HistBench, the first comprehensive benchmark for evaluating historical reasoning in LLMs. HistBench comprises 414 expert-curated questions spanning diverse time periods, world regions, and 29 languages. The tasks include both factual retrieval and interpretive analysis across texts, manuscripts, and images. Crucially, the questions are stratified into three difficulty levels, not by model performance but by domain experts (professional historians), who assess the inherent research difficulty of the tasks. This classification follows six structured criteria: rarity of source knowledge, linguistic complexity, format heterogeneity, perceptual accessibility, interdisciplinary scope, and reasoning depth. By embedding the disciplinary judgment of historians into the benchmark design, HistBench resists shallow retrieval strategies and provides fine-grained diagnostics of LLMs' ability to engage with historical reasoning.

HistBench enables us to systematically evaluate how well LLMs and agents perform on historical reasoning tasks. However, current models show clear limitations on HistBench: they underperform on fragmented, multilingual, and multimodal sources, and no existing system is tailored to the unique challenges of historical research. In response, we propose HistAgent, a domain-specialized agent for history. HistAgent augments a strong LLM with modular tools to meet the epistemic and technical demands of historical inquiry, including OCR for handwritten manuscripts (e.g., Transkribus, Asian-script OCR), multilingual translation with provenance tracking, reverse image search for historical visuals, and scholarly literature retrieval. HistAgent spans text, image, audio, manuscript, and video, applying source-aware workflows for extraction, parsing, and reasoning. On HistBench, HistAgent substantially outperforms base LLMs and generalist agents, while maintaining strong

performance on general benchmarks such as GAIA.

In summary, this paper makes the following contributions:

- **HistBench**: the first large-scale and comprehensive benchmark involving 414 high-quality, expert-reviewed, multimodal, multilingual questions for evaluating historical reasoning in LLMs.

- **HistAgent**: domain-specialized agent integrating tools and mechanisms aligned with historical research, improving accuracy on challenging historical tasks while preserving general capability.

- **Comprehensive Empirical Validation**: extensive experiments showing current LLMs and agents fail on historical reasoning, and demonstrating HistAgent's clear advantage on HistBench and competitive performance on GAIA.

## 2. Related Work

**Generalist Agents.** Recent LLM-based agents have shown strong ability on open-ended tasks that require planning, tool use, and information seeking. GAIA (Mialon et al., 2023) evaluates general assistant-like systems on real-world questions that are difficult for LLMs without tools. Along this direction, AutoAgent (Tang et al., 2025) constructs executable multi-agent workflows from natural-language descriptions, while OWL (Hu et al., 2025) uses an Orchestrator–Worker design to assign subtasks to specialist agents. Deep Research systems further combine web browsing, document reading, and grounded synthesis for long-form information tasks (OpenAI, 2025; Roucher et al., 2025). These systems show the promise of tool-augmented agents, but they are designed for broad task solving rather than the specific needs of historical research, such as multilingual source handling, evidence comparison, and multimodal interpretation.

**Domain-specific Agents and Benchmarks.** A growing line of work studies domain-specific agents and evaluations for fields that require expert knowledge. For example, Chem-Crow (Bran et al., 2023) integrates chemistry tools for syn-

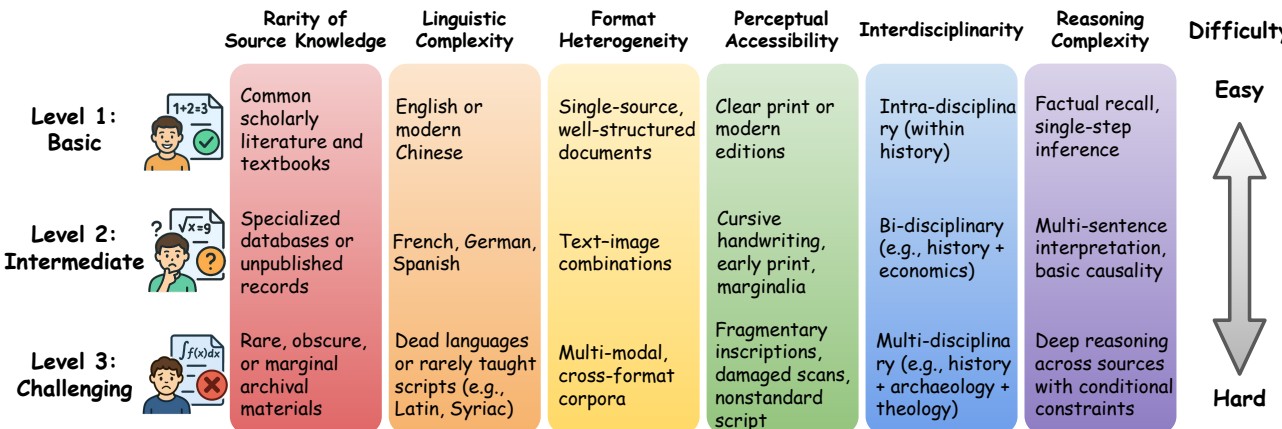

*Figure 3.* Difficulty level definitions across six structured evaluation dimensions.

thesis and materials-related tasks, CRISPR-GPT (Huang et al., 2024) supports gene-editing experiment design, and DS-Agent (Guo et al., 2024) uses case-based reasoning for automated data science. In parallel, domain-specific benchmarks have been proposed for software engineering, medicine, law, science, and mathematics, such as SWE-Bench (Jimenez et al., 2023), PubMedQA (Jin et al., 2019), LegalBench (Guha et al., 2023), SciBench (Wang et al., 2023), and PHYBench (Qiu et al., 2025). These studies suggest that broad benchmarks often miss domain-level failure modes and that specialized evaluation is needed when tasks depend on field-specific evidence, concepts, and reasoning practices.

**History-oriented Evaluation.** Existing history-oriented evaluations remain limited for testing historical research ability. HiST-LLM (Hauser et al., 2024) evaluates expert historical knowledge based on the Seshat Global History Databank, while KE-MHISTO (Graciotti et al., 2025) evaluates long-tail historical knowledge in a multilingual music-history setting. HLE (Phan et al., 2025) includes challenging history questions, but only as a small subset of a broader expert benchmark. Recent work also discusses how LLMs may assist historians with source-based research (Gonzalez Garcia & Weilbach, 2023) and calls for historical language models adapted to historical corpora (Varnum et al., 2024). However, existing benchmarks still lack systematic coverage of multilingual sources, multimodal evidence, source retrieval, and evidence-grounded reasoning, which motivates HistBench and HistAgent.

## 3. HistBench Benchmark

### 3.1. Overview

As there has been no dedicated benchmark for historical research and humanities reasoning, **HistBench** is the first benchmark specifically designed to evaluate LLM capabil-

ities in historical research and humanities reasoning. It contains 414 human-authored questions by over 40 contributors (students, graduate researchers, and domain experts), spanning two formats (exact match and multiple choice), six evaluation dimensions (e.g., source processing, interdisciplinary synthesis), and three difficulty levels. Each question is built from authentic historical materials with rich metadata, including source type, reasoning skill, and thematic category.

Unlike existing resources (Table 1), which are mainly history-related subsets of general-purpose QA benchmarks or rely on synthetic question generation (e.g., HiST-LLM (Hauser et al., 2024)), HistBench uniquely combines (1) human curation, (2) high multimodality (75% non-text items: texts, manuscripts, images, audio/video, inscriptions), and (3) discipline-grounded design across 29 ancient and modern languages, 36 subfields, and 20+ geographic regions. This provides a rigorous and realistic setting for evaluating historical reasoning beyond factual recall.

### 3.2. Benchmark Curation

All questions in HistBench are newly authored or adapted from authentic historical materials, designed to evaluate reasoning competencies central to professional historical research. More than 40 contributors, including undergraduate students, graduate researchers, and senior scholars, participate in drafting, refining, and validating the dataset. The overall curation pipeline consists of two main stages: question formulation and quality control.

**Question Formulation.** Our data include a wide range of sources, including manuscripts, inscriptions, early printed texts, archival records, visual artifacts, and audio/visual materials. Each question is constructed within a standardized template that requires the following fields: problem statement, expected answer, detailed explanation, difficulty level,

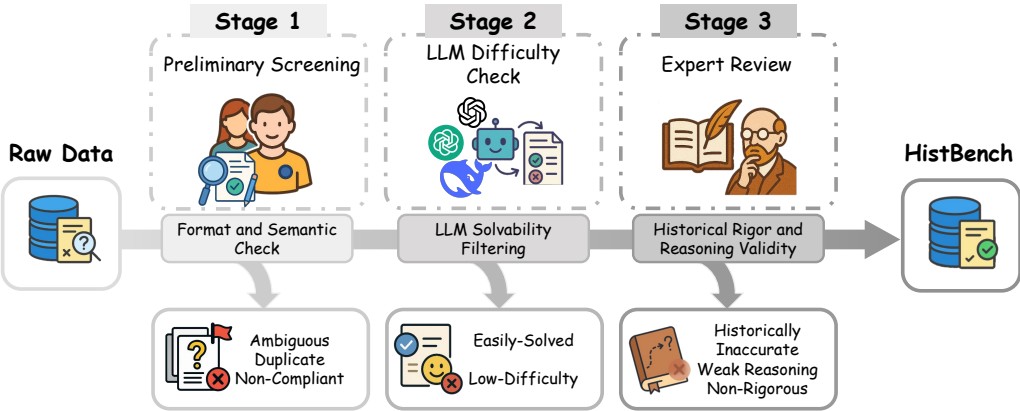

*Figure 4.* Multi-Stage Question Review Pipeline for HistBench

reasoning dimension tags, and bibliographic references (refer to Appendix B.1 for more details). This format ensures consistency and traceability across the dataset.

To ensure both precision and interpretive breadth, two complementary formats are adopted:

- **Exact-Match**: fact-oriented tasks requiring precise responses such as names, dates, or locations, to test precision in historical retrieval and reasoning.

- **Multiple-Choice**: tasks with one correct option and three distractors that reflect common misconceptions or alternative interpretations. This format prompts multi-angle reasoning, enabling quasi-open-ended evaluation while preserving reproducibility and scalability without heavy reliance on expert grading.

To support layered evaluation, all questions are stratified into three difficulty levels: Level 1 (Basic), Level 2 (Intermediate), and Level 3 (Challenging). Difficulty is defined according to a structured rubric of six academic dimensions: (1) rarity of source knowledge, (2) linguistic complexity, (3) format heterogeneity, (4) perceptual accessibility, (5) interdisciplinary scope, and (6) reasoning complexity (see Fig. 3). Importantly, this definition follows the perspective of human historians rather than that of current LLMs. Authors are assigned tasks in accordance with their expertise: trained research assistants draft Level 1 questions, graduate researchers contribute Level 2 ones, and university faculty or senior scholars author the most challenging items. Representative examples are provided in Appendix B.6.

**Quality Control.** Following the formulation, all questions experience a rigorous three-stage review process, shown in Fig. 4:

1. **Preliminary Screening**: members of the project team examine questions for clarity, completeness, and redun-

dancy, returning ambiguous or poorly defined items for revision.

2. **LLM Difficulty Check**: each candidate question is tested against GPT-4o, GPT-4-mini, and DeepSeek-R1. A question is discarded only if at least two models answer it correctly without access to supporting sources. This filtering reduces trivial items while mitigating model-specific bias.

3. **Expert Validation**: historians review the surviving questions for evidentiary grounding, methodological rigor, and interpretive validity. Items that fail to meet academic standards are either revised or removed.

Through this pipeline, a larger initial pool of candidate items is refined into the final set of 414 high-quality questions. Importantly, some retained items remain solvable by strong models such as GPT-4o, demonstrating that HistBench is not adversarially constructed but instead reflects the realistic challenges faced in historical research. The curation process thus balances academic rigor, discriminative power, and diversity, providing a reliable foundation for evaluating LLMs on history-focused reasoning tasks.

### 3.3. Dataset Characteristics and Coverage

**Quantitative Overview.** HistBench is initially constructed with 1034 questions. After the first round of screening, 720 are retained, and following combined LLM-based filtering and expert review, 414 high-quality items remained. Of these, 306 are exact-match and 108 are multiple-choice. The predominance of exact-match questions highlights the benchmark's focus on fine-grained factual precision, while the multiple-choice format captures interpretive discrimination and reasoning across structured alternatives.

**Language Diversity.** As illustrated in Figure 2b, questions span 29 modern and ancient languages, including

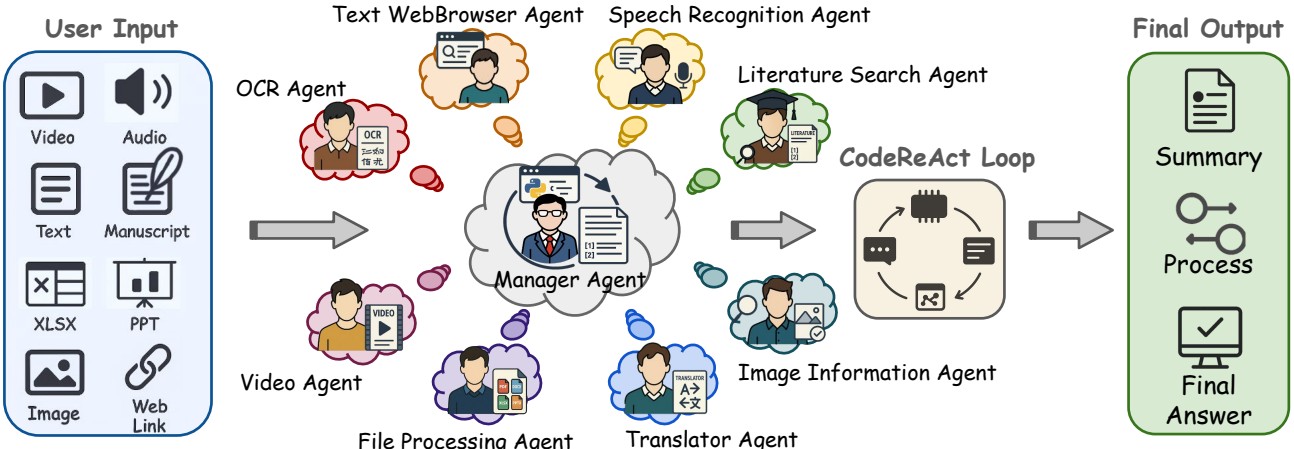

*Figure 5.* The architecture of HistAgent, an agent for historical reasoning. The system takes multimodal user inputs (e.g., text, audio, video, images, documents) and coordinates specialized agents through a centralized Manager Agent. Each agent is responsible for dedicated tasks such as OCR, speech recognition, literature search, translation, and file processing. The Manager Agent orchestrates these agents within a CodeAct loop to iteratively refine reasoning and evidence validation, ultimately generating summarized, processed, and fully cited academic answers.

widely used academic languages such as English, Chinese, French, and Russian, as well as historically significant but low-resource languages such as Classical Chinese, Latin, Syriac, and Tibetan. This multilingual coverage highlights a key limitation of current LLMs, many of which fail even simple tasks in low-resource languages. Though some languages are represented by only a handful of questions, even a single carefully designed item can expose critical weaknesses.

**Source Modalities.** HistBench incorporates diverse source types, including texts, manuscripts, inscriptions, images, and audio/visual recordings. Many items involve fragmentary, cursive, or otherwise non-standard formats that stress-test OCR systems and multimodal reasoning pipelines. By encompassing both clean and noisy materials, the benchmark evaluates the robustness of LLMs in realistic scholarly contexts.

**Domain and Geographic Coverage.** Histbench covers 36 historical subfields and more than 20 geographic regions (Fig. 2c). The domains include political and cultural history, material culture, science and medicine, and environmental change. Geographic coverage spans all inhabited continents, from East Asia and Europe to Latin America and Africa, with deliberate inclusion of underrepresented areas such as papyrology and Siberian ethnography.

**Temporal Coverage.** HistBench covers the full temporal scope of human history, following a five-part periodization widely used in global historiography: Prehistory, Ancient History, the Middle Ages, Modern History, and Contemporary History (Woolf, 2011; Arnold, 2000). This long-range

design enables systematic assessment of historical reasoning across distinct epistemic contexts and source traditions.

## 4. HistAgent

**HistAgent** is a domain-specialized agent system designed to enhance historical reasoning by augmenting a strong LLM with task-critical tools and coordinated sub-agents (Fig. 5). Unlike basic retrieval systems, HistAgent supports academic search, multimodal inputs (texts, manuscripts, images, audio/video), and produces fully cited responses grounded in primary and secondary sources.

### 4.1. Architecture

HistAgent follows a manager–specialist design. A central **Manager Agent** decomposes queries, dispatches subtasks, verifies evidence, and runs an iterative CodeAct loop to invoke specialized agents and tools. In each iteration, the manager agent emits a Python code snippet that calls sub-agent functions or tools; the snippet executes in a secure sandbox and writes its output to shared memory. The agent then validates each result (checking exact quotes and bibliographic data) and repeats until task completion criteria are met. In the final step, the agent synthesizes all validated outputs into a structured, citation-complete response. Each **specialist agent** targets a concrete challenge in historical reasoning: web and literature retrieval for scholarly sources, OCR for manuscripts and early prints, translation for low-resource and historical languages, and multimodal analysis for images, audio, or video. The Manager validates intermediate outputs (e.g., quotes and bibliographic data) and integrates them into a single, citation-complete answer.

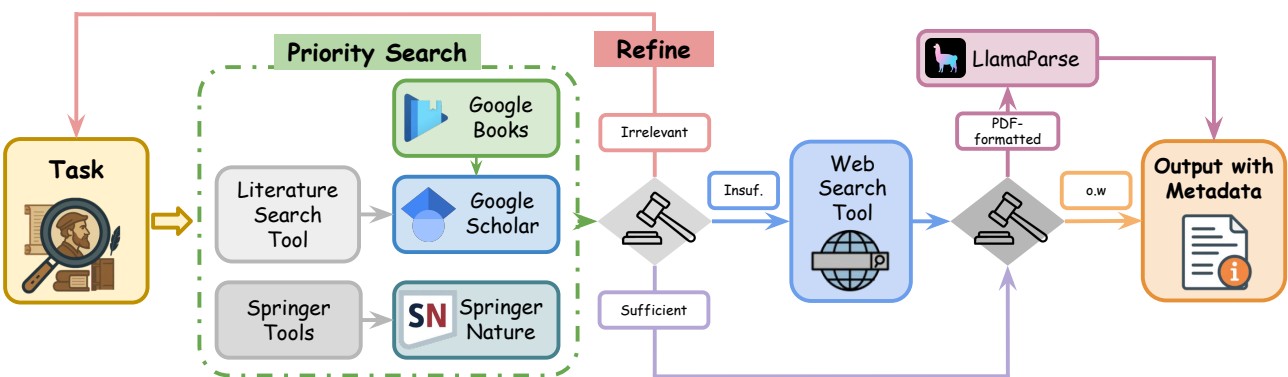

*Figure 6.* Literature Search Agent is an agent specifically designed for academic searching. Its core components include a search strategy module, integration with web search tools, and a ReAct loop.

An overview of specialist agents and their toolkits is shown in Table 2, with full configuration details in Appendix C.

### 4.2. Key Specialized Component: Literature Search Agent

The Literature Search Agent is a highly specialized component within the HistAgent framework, engineered as a ReAct-based agent driven by a large language model (LLM). Its core architectural design focuses on emulating a meticulous academic research process through a structured, multi-stage interaction with a curated set of web-based tools, primarily implemented through browser-use (Browser Use contributors, 2025), which can connect the agent to the real Chrome browser instance. This setup allows the agent to operate within the user's active browser profile, inheriting authentication states and personalized settings for seamless access to academic resources.

Its internal architecture is centered around a protocol-driven retrieval engine that ensures the reliability, interpretability, and academic integrity of the information it gathers. The agent **prioritizes scholarly sources** such as Google Scholar and Springer Nature, employs a **modular toolkit** for interfacing with both academic databases and general-purpose web sources, and allows users to configure search depth and behavior via **interpretable parameters**. Retrieved content is enriched with bibliographic metadata, quoted evidence, and **full traceability** links to support verification and integration into downstream workflows. For more details, please refer to Appendix D.

## 5. Experiment

### 5.1. Experiment Setup and Evaluation

**Datasets.** We evaluate HistAgent on three benchmarks. (1) **HistBench**: our 414-question benchmark covering multiple modalities, 29 languages, and three difficulty levels. (2) **GAIA** (Mialon et al., 2023): 165 real-world tasks for testing generalization. (3) **HLE-History** (Phan et al., 2025): a 56-question subset of Humanity's Last Exam focused on history.

*Table 2.* Function Overview of HistAgent Specialist Agents.

| Agent | Modality | Core Tools |
|---|---|---|
| Text WebBrowser | Text | Multi-step web search and page parsing (`LocalGoogleSearchTool`, `VisitTool`). |
| Image Information | Image | Reverse image search and provenance analysis (`GoogleLensSearchTool` with SSIM filtering). |
| Literature Search | Scholarly text | Peer-review retrieval and PDF parsing (Scholar websites, `SpringerDownloadAndParseTool`). |
| File Processing | Documents | Typed file parsing: `PDFTool`, `DOCXTool`, `XLSXTool`, `PPTXTool`. |
| OCR | Image text | Manuscript transcription (Transkribus, Asian-script OCR). |
| Speech Recognition | Audio | Whisper-based transcription with LLM correction. |
| Translator | Multilingual text | Bidirectional translation with provenance preservation. |
| Video | Video | Frame extraction (yt-dlp, OpenCV). |

*Table 3.* Performance accuracy (%) on HistBench. **Note:** All standalone large language models are used **with web search** capabilities. The best value in each column is highlighted with  darker blue , and the second-best score with  lighter blue .

| Agent/Model | | Level 1 | Level 2 | Level 3 | Average |
|---|---|---|---|---|---|
| HistAgent (GPT-4o) | pass@1 | 28.92 | 26.16 | 32.89 | 28.50 |
| | pass@2 | 36.14 | 35.47 | 39.47 | 36.47 |
| ODR-smolagents (GPT-4o) | pass@1 | 16.27 | 21.51 | 22.37 | 19.57 |
| | pass@2 | 20.48 | 28.49 | 27.63 | 25.12 |
| DeepSeek-R1:online | | 11.45 | 19.77 | 11.84 | 14.98 |
| GPT-4o:online | | 13.86 | 21.51 | 22.37 | 18.60 |
| o4-mini-high:online | | 28.92 | 30.81 | 31.58 | 30.19 |
| Grok 3:online | | 13.25 | 19.77 | 22.37 | 17.63 |

**Baselines.** We compare HistAgent with open Deep Research (ODR-smolagents) (Roucher et al., 2025), an open-source reproduction of Deep Research, under identical conditions and using the same base model (GPT-4o) to isolate agent design effects. We also report results for strong general LLMs with search (DeepSeek-R1 (Guo et al., 2025), GPT-4o (Hurst et al., 2024), o4-mini-high (OpenAI, 2025), Grok 3 (xAI, 2025)). Each question is run in a fresh session with a fixed budget of tool call.

**Evaluation Metrics.** HistBench accuracy is defined as the percentage of questions marked "Correct" by both an LLM judge and human expert sampling (100 cases). GAIA and HLE-History follow their official scoring protocols: GAIA uses exact match with normalization; HLE-History applies the official LLM-judge pipeline. Responses follow a structured output format with final answer and reasoning summary.

Further details are in Appendix H.2 and H.3.

### 5.2. Results

**HistBench**   Table 3 reports the level-wise and average accuracies on HistBench. HistAgent (GPT-4o) attains 28.50% pass@1 and 36.47% pass@2 on average. The corresponding averages for ODR-smolagents (GPT-4o) are 19.57% pass@1 and 25.12% pass@2, and for GPT-4o with web search are 18.60% (single-row setting in Table 3). Among the standalone models with web search, the best average pass@1 result is 30.19% from o4-mini-high:online.

*Table 5.* Performance on the HLE History Subset (56 Questions, LLM Judged, %). To ensure fair comparison, both HistAgent and ODR-smolagents are based on GPT-4o.

| System | Pass@1 | Pass@2 | Pass@3 |
|---|---|---|---|
| HistAgent (GPT-4o) | **28.57** | **39.29** | **42.86** |
| ODR-smolagents (GPT-4o) | 17.86 | 25.00 | 28.57 |
| GPT-4o + web search | 8.93 | 19.64 | 25.00 |

**HLE History Subset**   On the 56-question HLE History Subset (Table 5), all systems understandably struggle on many of these prompts. HistAgent (GPT-4o) yields 28.57% pass@1, 39.29% pass@2, and 42.86% pass@3. In comparison, ODR-smolagents (GPT-4o) yields 17.86%, 25.00%, and 28.57%, respectively, while GPT-4o with web search reaches 8.93%, 19.64%, and 25.00%. HistAgent therefore answers 60% more questions correctly at pass@1 than the ODR-smolagents baseline and more than triples the pass@1 performance of the direct GPT-4o baseline.

**GAIA Validation**   On the 165-question GAIA validation split (Table 4), HistAgent (Claude–3.7–sonnet) answers 99 questions correctly, corresponding to 60.00% pass@1. ODR-smolagents (o1) records 55.15% pass@1 on the same split. The level-wise accuracies for HistAgent are 69.81% (Level 1), 61.63% (Level 2), and 34.62% (Level 3); the corresponding values for ODR-smolagents are 67.92%, 53.49%, and 34.62%.

*Table 4.* Performance accuracy (%) on the GAIA validation set (pass@1)

| Agent | Model | Average | Level 1 | Level 2 | Level 3 |
|---|---|---|---|---|---|
| HistAgent | Claude–3.7–sonnet | **60.00** | 69.81 | 61.63 | 34.62 |
| ODR-smolagents | o1 | 55.15 | 67.92 | 53.49 | 34.62 |

*Table 6.* HistBench accuracy (%) grouped by question modality. Text-only questions contain only textual information, while multimodal questions include additional attachments such as images, presentation slides, handwritten manuscripts, or other documents.

| Difficulty Level | HistAgent (GPT-4o) | | ODR-smolagents (GPT-4o) | | LLMs with Search | | | |
|---|---|---|---|---|---|---|---|---|
| | pass@1 | pass@2 | pass@1 | pass@2 | Deepseek-R1 | GPT-4o | o4-mini-high | Grok3 |
| **Text-only Questions** | | | | | | | | |
| Level 1 | 21.52 | 29.11 | 21.52 | 25.32 | 5.06 | 13.92 | **40.51** | 15.19 |
| Level 2 | 18.42 | 28.95 | 12.66 | 28.95 | 21.05 | 21.05 | **44.74** | 15.79 |
| Level 3 | 12.50 | 12.50 | 12.50 | 12.50 | 25.00 | 12.50 | **62.50** | 25.00 |
| **Multimodal Questions** | | | | | | | | |
| Level 1 | 35.63 | **42.53** | 11.49 | 16.09 | 17.24 | 13.79 | 18.39 | 11.49 |
| Level 2 | 28.36 | **37.31** | 20.15 | 28.36 | 19.40 | 21.64 | 26.87 | 20.90 |
| Level 3 | 35.29 | **42.65** | 23.53 | 29.41 | 10.29 | 23.53 | 27.94 | 22.06 |

## 5.3. Analysis

**HistBench.** On HistBench, HistAgent achieves the strongest overall agent performance, with 28.5% pass@1 and 36.47% pass@2. Under the matched GPT-4o backbone setting, it improves pass@2 by 11.35 points over the open-source ODR-smolagents, while the gain of ODR-smolagents over GPT-4o with web search remains modest. This shows that historical tasks require more than generic agent tuning. HistAgent gains from history-oriented modules such as handwriting OCR, multimodal analysis, translation, and scholar search with query optimization.

Despite using the smaller GPT-4o backbone, HistAgent remains competitive with stronger closed-source models such as o4-mini-high, suggesting that targeted tool and reasoning design can partly offset model scale and reduce inference cost. In future work, we will pair HistAgent with stronger foundation models, such as o4-mini-high, to test the upper bound of this tool-augmented reasoning pipeline.

**HLE History Subset.** On the HLE History Subset, HistAgent also consistently outperforms the baselines, with a clear pass@2 lead over ODR-smolagents. Since HLE covers broader historical topics and uses more open-ended questions, this result suggests that HistAgent generalizes beyond HistBench. Its gains come from history-oriented modules for multimodal analysis, OCR, translation, and scholar search, which help retrieve and verify evidence from unstructured images, ambiguous handwriting, cross-lingual references, and incomplete context. By contrast, ODR-smolagents relies on more generic search and weaker query optimization, leading to limited gains on these tasks. These findings confirm that HistAgent's architecture is not over-fitted to HistBench but remains effective and robust on an external history-focused benchmark.

**GAIA Validation.** HistAgent outperforms GPT-4o with web search and ODR-smolagents on GAIA, demonstrating

that its reasoning framework and tool set generalize beyond history-specific tasks. This suggests that task decomposition, OCR, translation, and scholar search also support evidence retrieval and verification in broader humanities and social science settings.

**Modality-based Performance Analysis.** The modality-based breakdown in Table 6 shows that one of HistAgent's main advantages lies in multimodal reasoning. While strong search-enabled LLMs remain competitive on text-only questions, HistAgent consistently outperforms ODR-smolagents and standalone baselines on multimodal questions. This suggests that its gains come from processing attached evidence, such as images, manuscripts, slides, and documents, through OCR, translation, and multimodal analysis. Since the backbone is fixed, the results indicate that the agent framework itself drives much of the improvement.

**Qualitative Insights.** A qualitative inspection of successful HistBench cases suggests that tool usage is often decisive. In particular, OCR for handwritten or inscribed text, translation of historical languages, and scholar-oriented retrieval frequently serve as key intermediate steps for resolving historical evidence. These observations are consistent with the improvements observed on multimodal questions and the overall performance gains on HistBench and HLE.

## 6. Conclusion

We introduce HistBench, a 414-question benchmark for historical reasoning, and HistAgent, an agent that aligns tools and workflows with core tasks in history. HistBench spans texts, manuscripts, images, audio/video, and inscriptions, covers 29 languages, and uses an expert rubric to label difficulty. On this benchmark, HistAgent (GPT-4o) reaches 28.5% pass@1 and 36.47% pass@2, outperforming general LLMs with web search and an open reproduction of Deep Research under matched settings.

## Impact Statement

This paper introduces HistBench and HistAgent to improve the evaluation and support of historical reasoning in AI systems. By focusing on multilingual and multimodal historical sources, this work may help researchers, educators, and students better access and analyze historical materials, especially those written in low-resource languages or preserved in non-textual formats. At the same time, AI-generated historical outputs may contain errors, bias, or incomplete interpretations, and should not be used without expert verification. We emphasize that HistAgent is intended to assist human researchers rather than replace scholarly judgment, and responsible use requires critical review and domain expertise.

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

# A. Related Works

## A.1. Generalist Agent

Generalist agent is for solving general real-world tasks such as GAIA(Mialon et al., 2023). OMNE (Jiang et al., 2024) introduces a column-structured long-term memory that agents update during inference to refine policies without retraining. AutoAgent (Tang et al., 2025) compiles natural-language workflow descriptions into executable multi-agent pipelines. OWL (Hu et al., 2025) adds structured orchestration using a CAMEL-based Orchestrator–Worker pattern, where the orchestrator delegates subtasks to specialists via explicit transfer actions. For information-dense tasks, OpenAI Deep Research (OpenAI, 2025) combines web browsing, document parsing, and grounded synthesis to produce cited reports, while open Deep Research by Smolagents (Roucher et al., 2025) reproduces the same workflow in the open-source domain using a Python-driven CodeAgent that reduces communication overhead. They jointly define today's design space for scalable generalist multi-agent solutions.

## A.2. Domain-specific Agent

Domain-specific agents address the limitations of general-purpose LLMs in tasks requiring deep expertise. A primary approach to imbue such specificity is parametric adaptation through fine-tuning on domain-centric corpora, enabling models to learn specialized terminology, data patterns, and reasoning pathways. Notable examples include CRISPR-GPT (Huang et al., 2024), an agent designed to automate and enhance the design of CRISPR-based gene-editing experiments by leveraging domain knowledge and external tools, BrainGPT (Li et al., 2025), a model clinically visual instruction-tuned for 3D brain CT radiology report generation , and DeepSeekMath (Shao et al., 2024), which demonstrates advanced mathematical reasoning after extensive training on mathematical texts. These methodologies are crucial for creating agents that are not only knowledgeable but also more reliable and effective within their specific operational contexts.

Beyond foundational knowledge embedding, the efficacy of domain-specific agents is significantly amplified by their ability to interact with and act upon their environment through tool integration. ChemCrow (Bran et al., 2023), for instance, leverages a suite of 18 expert-designed chemistry tools to perform complex tasks in organic synthesis, drug discovery, and materials design, showcasing enhanced performance in chemistry-related problem-solving. Other advanced approaches include Case-Based Reasoning (CBR), enabling agents such as DS-Agent (Guo et al., 2024) to learn from past expert solutions in fields like automated data science. These approaches collectively aim to produce agents that are not only competent in their domain but also robust and trustworthy in real-world settings.

## A.3. Domain-specific Benchmarks

The evaluation of large language models (LLMs) is moving from broad assessments toward domain-specific benchmarks (DSBs). General-purpose tests often miss the detailed strengths and weaknesses of LLMs when applied in real-world settings. Common issues with standard benchmarks include limited coverage of field knowledge, poor alignment with practical tasks, vulnerability to data contamination that encourages memorization, and simple question formats that do not test multi-step or advanced reasoning (Wang et al., 2023; Alaa et al., 2025; Hong et al., 2025; Bodensohn et al., 2025). Questions about data quality and contamination have led to efforts such as HLE, which invests in carefully curated, contamination-resistant items (Phan et al., 2025). As a result, the number of DSBs has grown rapidly. These new tests help guide model improvements, set realistic expectations for deployment, and pinpoint specific model capabilities in different areas (Guha et al., 2023; Pipitone & Alami, 2024).

This trend towards specialization is evident across numerous domains. In software engineering, benchmarks like SWE-Bench (Jimenez et al., 2023) evaluate LLMs on resolving real-world GitHub issues, moving beyond simpler code generation tasks assessed by benchmarks like HumanEval or MBPP (Chen et al., 2021; Austin et al., 2021; Jimenez et al., 2023; Phan et al., 2025). The medical domain utilizes benchmarks such as PubMedQA (Jin et al., 2019) and the MultiMedQA suite (Singhal et al., 2023) to assess medical knowledge and reasoning, although ongoing research seeks to improve alignment with clinical practice and address benchmark saturation (Singhal et al., 2023; Alaa et al., 2025). Legal AI evaluation employs benchmarks like LegalBench (Guha et al., 2023) for diverse legal reasoning tasks and specialized versions like LegalBench-RAG (Pipitone & Alami, 2024) focusing on retrieval-augmented generation crucial for fact-intensive legal work. For scientific and mathematical reasoning, SciBench (Wang et al., 2023) presents collegiate-level problems , RV-Bench (Hong et al., 2025) targets genuine mathematical understanding , and PHYBench (Qiu et al., 2025) specifically assesses complex physics reasoning using problems from global exams and competitions. Finance has seen the development of

benchmarks like FinanceQA (Mateega et al., 2025), tailored to evaluate performance on tasks mirroring real-world financial analysis. Collectively, these domain-specific evaluations are crucial for probing deeper LLM capabilities and fostering the development of models suitable for specialized professional deployment.

### A.4. History Benchmarks

In the domain of history, based on a subset of the Seshat Global History Databank, HiST-LLM evaluates the possession of expert historical knowledge of Seven Models. (Hauser et al., 2024) However,there is a discrepancy between historical knowledge and historical research.HLE can be regarded as a combination of many domain-specific benchmarks. In terms of the historical questions in HLE, although the capabilities of LLMs in historical research is indeed evaluated (as opposed to the benchmarks mainly focused on historical knowledge), the total number of questions is only 56. (Phan et al., 2025) For existing benchmarks, there are limitations in both the scope of the questions and the disciplinary characteristics.

We need to detail several limitations in these benchmarks. The creators of Gaia (Mialon et al., 2023) indicate that Gaia has three limitations—Missing evaluations: Shortage of evaluation of trace leading to the answer; On the cost of designing unambiguous questions: Shortage of ambiguous questions that may appear in the daily usage scenario; Lack of linguistic and cultural diversity: Shortage of questions in languages other than standard English.The creators of HiST-LLM also indicate the limitations of the database they use (the subset of the Seshat Gloval History Databank). First, their data are mostly from sources in English;Second, the expertise and background of the research assistants may influence the definition of variables. Third, the benchmark is only the reflection of the current recognition. In addition to these limitations, it is necessary to repeat that HiST-LLM is a benchmark designed for the evaluation of the possession of historical knowledge instead of the capabilities of historical research such as literature retrieval, historical source retrieval, historical analysis, historical source processing, and interdisciplinary.

Compared to the comprehensive benchmark like HLE, an independent domain-specific benchmark can be designed more suitable for a specific domain without the requirement of consistency and inclusiveness of different domains. To evaluate the capabilities of LLMs in the domain of historical research more adequately, creating a larger and more systemic dataset is necessary. If we take historical questions in HLE as an independent historical benchmark, there will be more space for improvement. First, the temporal and spatial scope of topics should be expanded with more questions. Second, the types and depth of functions to be evaluated should be richer. For example, we cannot find historical materials in the form of audios or video in historical questions of HLE. Third, in the context of the history discipline, the questions can be divided further in different groups according to their difficulty levels, thematic categories and evaluation criteria, which helps to constructing a more diverse but clear database.

Considering these limitations, there remains substantial room to design independent domain-specific benchmarks for historical research. Recent work also shows that historians are already exploring how LLMs might assist research tasks beyond factual recall (Gonzalez Garcia & Weilbach, 2023), underscoring the need for more rigorous and larger-scale evaluation resources. In parallel, there are calls for training or adapting LLMs specifically on historical corpora to better support such applications (Varnum et al., 2024). These observations further motivate HistBench as a systematic and scalable benchmark for historical reasoning.

## B. HistBench Benchmark

An example of the HistBench data format is provided in `https://anonymous.4open.science/r/HistBench-8B86`.

### B.1. Submission Template Format

To illustrate how these elements are applied in practice, Table 7 provides two sample entries from different difficulty levels.

- **(a) Difficulty Level:** Assigned as Level 1, 2, or 3 based on rubric criteria (see Section 4.3).

- **(b) Question Prompt:** A clear and concise question targeting a specific historical issue requiring domain expertise.

- **(c) Required Data:** Source materials referenced or used (e.g., documents, images, audio/video).

- **(d) Answer:** A definitive, validated response—either as a selected option or short text span.

- **(e) Answer Explanation:** A concise justification based on evidence and reasoning.

- **(f) Source References:** URLs or bibliographic citations supporting the answer.

- **(g) Topic/Methodology:** Thematic or methodological classification (e.g., diplomatic history, material culture).

- **(h) Contributor Name:** Full name of the author.

- **(i) Contributor Affiliation:** Institutional affiliation at the time of contribution.

This format ensured each question adhered to standards of academic transparency and could be independently reviewed and validated.

| Field | Q001 (Level 1) | Q002 (Level 3) |
|---|---|---|
| ID | Q001 | Q002 |
| Answer Type | Multiple Choice | Exact Match |
| Question | What year did the Treaty of Westphalia end the Thirty Years' War? | Translate and date the following Latin inscription found on a Roman milestone in Gaul. |
| Data Requirements | Modern European history textbook excerpt | Image of milestone inscription (Latin) |
| Answer | 1648 | Circa 220 CE |
| Answer Explanation | The Treaty of Westphalia was signed in 1648, ending the Thirty Years' War in Europe. | The inscription, typical of the Severan dynasty period, was dated to around 220 CE using epigraphic style. |
| Source Materials | *Merriman, A History of Modern Europe*, p. 203 | *Corpus Inscriptionum Latinarum*, Vol. XIII |
| Thematic Category | Political History | Epigraphy / Classical Studies |
| Evaluation Criteria | Factual recall | Source processing; temporal inference; Latin translation |
| Contributor's Name | XX | XXX |
| Contributor's Affiliation | XXX | XXX |

*Table 7.* Vertical Format Sample Entries from HistBench

**B.2. Evaluation Dimensions**

Each question in HistBench is tagged with one or more reasoning dimensions that reflect core competencies required in historical research. These dimensions are designed to evaluate a model's ability to retrieve, interpret, and reason about diverse historical materials across linguistic, modal, and disciplinary boundaries. A detailed taxonomy of the six evaluation dimensions is provided in Table 8.

| ID | Dimension | Description |
|----|-----------|-------------|
| 1 | Bibliographic Retrieval | The capability to locate information embedded in scholarly texts, monographs, or journal articles using digital or library-based search strategies. |
| 2 | Source Identification | The capability to recognize or locate specific historical sources, including manuscripts, digitized archives, or visual primary materials. |
| 3 | Source Processing | The capability to extract and interpret information from non-textual formats such as handwritten documents, historical images, audio, or video. |
| 4 | Historical Analysis | The capability to engage in historically grounded reasoning, including causal inference, ideological analysis, and interpretation of events or institutions. |
| 5 | Interdisciplinary Integration | The capability to draw upon methods and frameworks from adjacent disciplines (e.g., archaeology, linguistics, religious studies) to support historical understanding. |
| 6 | Cultural Contextualization | The capability to interpret cultural cues, metaphors, sentiment, and identity markers within historically situated discourse. |

*Table 8.* Evaluation dimensions for historical reasoning tasks in HistBench.

### B.3. Language Distribution in HistBench

Table 9 provides the full list of languages represented in the HistBench dataset, along with their frequencies. These include both modern languages and historical scripts, reflecting the multilingual nature of historical research tasks. This diversity supports the evaluation of cross-lingual capabilities in AI systems, including translation, OCR, and historical reasoning across languages.

| Language | Count |
| --- | --- |
| English | 228 |
| Chinese | 52 |
| Russian | 22 |
| Japanese | 13 |
| French | 10 |
| Latin | 8 |
| German | 8 |
| Classical Chinese | 47 |
| Dutch | 5 |
| Tibetan | 2 |
| Armenian | 2 |
| Arabic | 2 |
| Khitan | 2 |
| Ancient Greek | 2 |
| Khmer | 1 |
| Indonesian | 1 |
| Old Tibetan | 1 |
| Sanskrit | 1 |
| Old Uyghur | 1 |
| Middle Polish | 1 |
| Aramaic | 1 |
| Danish | 1 |
| Bosnian | 1 |
| Italian | 1 |
| Macedonian | 1 |
| Yukaghir | 1 |

*Table 9.* Languages represented in HistBench questions.

### B.4. Temporal distribution of questions

Table 10 summarizes the distribution of questions across historical periods.

| Historical Period | Number of Questions |
| --- | --- |
| Ancient History (to ~500 CE) | 90 |
| Medieval History (500–1500) | 85 |
| Early Modern History (1500–1800) | 95 |
| Modern History (1800–1945) | 80 |
| Contemporary History (1945–present) | 64 |

*Table 10.* Chronological coverage of questions in HistBench.

### B.5. Domain Taxonomy in HistBench

To support comprehensive evaluation of historical reasoning, HistBench includes tasks across 36 historical subdomains. These domains are grouped into eight overarching thematic categories as follows:

- **Political, Social, and Cultural History** — including diplomatic history, gender studies, intellectual history, and identity politics.

- **Classics and Ancient Civilizations** — including Greco-Roman studies, philology, epigraphy, and early textual traditions.

- **Art and Visual Culture** — including art history, iconography, visual semiotics, and historical image interpretation.

- **Material Culture and Archaeology** — including artifact studies, cultural heritage reconstruction, and field archaeology.

- **Environmental and Climate History** — including studies of climate shifts, ecological change, and environmental governance.

- **History of Science and Medicine** — including botany, astronomy, early scientific institutions, and traditional medicine systems.

- **Economic and Institutional History** — including labor systems, taxation, administrative structures, and legal codes.

- **Interdisciplinary and Comparative Studies** — including global history, translation history, mythology, and civilizational entanglements.

This classification reflects the diversity of methodologies and source types employed in historical research and supports fine-grained analysis of LLM capabilities across disciplinary boundaries.

### B.6. Sample Questions Across Difficulty Levels

To illustrate the scope and structure of questions in HistBench, we present three annotated examples—one for each difficulty level. These samples highlight variation in required source processing, historical reasoning, and interdisciplinary complexity.

- **Level 1 (Basic)** – Source verification and factual retrieval

- **Level 2 (Intermediate)** – Text-image synthesis and temporal reasoning

- **Level 3 (Challenging)** – Multilingual, multimodal, and interdisciplinary integration

## Level_1: Retrieval of Rare Data

**Question:**

Which ancient greek papyrus is this excerpt below taken from:

κληδὼν δὲ κλίτα ὡς Πίτωνα τοξεύσας,

τῶι Ζεὺς δαιδουχεῖ παρ' ἡμέραν κατ' ἄγαν,

τῶι γᾶς ἐν βώλοις ξανθοὶ τίκτονται καρποί.

A.Papyrus of Vienna G29825

B.Berlin Papyrus 6870

C.Papyrus Oxyrhynchus 2436

D.Papyrus Oslo 1

## Level_2: Multilingual Analysis

**Question:**

Один профессор однажды сказал в интервью: «Только очень наивный инопланетянин мог бы подумать о землянах как о единой цивилизации. На самом деле он обнаружил бы по меньшей мере девять цивилизаций, в которых есть своя структура.» В частности, он отметил: «Евразийская цивилизация (Россия, Украина, Белоруссия)... Отличается своеобразием социальной психологии народов, образом жизни в труде, быте, досуге. Здесь сложилась особая алкогольная цивилизация с вековыми питейными традициями, невиданными ни на Западе, ни на Востоке. И особая, неслыханная нигде ругань. Невиданная нигде смесь самоотверженности и скандальности, западной культуры и восточного быта. Словом, Азия, но в европейском обличье.» Какой он профессор?

## Level_3: Augmented Comprehensive Ability

**Question:**

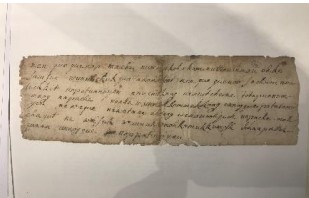
According to this deed from early eighteenth-century New England, what is the relationship between Johan Pattukqua and the sachem Quequenap (or Quequenab), who conveyed the land?

A. Johan Pattukqua was a witness at the meeting in which the deed was issued.
B. Johan Pattukqua was the scribe of the sachem.
C. Johan Pattukqua was a previous owner of part of the land.
D. Johan Pattukqua was the receiver of the land E. Johan Pattukqua was a friend of the sachem.

## Level_2: Better Multimodal Capability

**Question:**

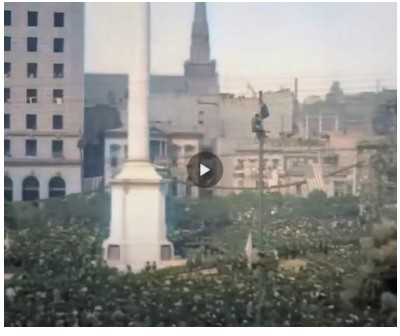

At 04:42 - 08:59, the screen shows a public green space. Which of the following historical facts about the place is incorrect?
A. There is a monument in the center of this saqure-like public space. This monument got its name from the pro-Union rallies held there on the eve of the Civil War.
B. When this clip was filmed, there was no underground garage in the area.
C. Based on the layout and facilities of the square in this clip, we can infer that the video was shot after the end of the Spanish-American War.
D. This place is on the list of California Historical Landmarks (CHL).

## Level_3: Debris Identification

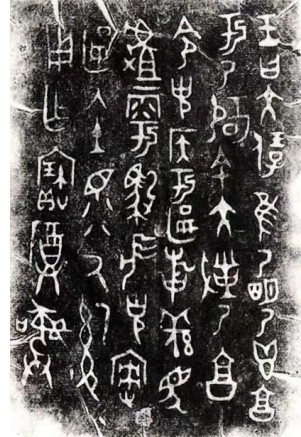

**Question:**

In all, how many vehicles and men were captured by Duo You（多友） in the attack on the Xian Yun（玁狁） tribe?

## Level_3: High-level Paleography

**Question:**

šrirnalandaram-ka  eltd[ilär] anta tägdökdä kamag [kuvrag] yıgılmıš ärdi, samt[so ačari] olar-nı birlä körüšü t[ükättök] dä š[ilaba]dre ačari bašınt[a][yegir]mi [a]čari-ka samtso ača]rig uduzturup [drmaguptake] ačari-ka yüküntür [gäli]… …dı-lar , d(a)rmaguptake [ačari] ärsär , kayken lüši [tegmä] šilabadre ačari käntü öz

Who is the samtso ačari mentioned in this Old Uighur text?
A.  Šilabaḍre          B. Faxiang          C. Xuanzang          D. Šingqu Säli          E. Aśoka

| Agent | Modality | Core Tools |
|---|---|---|
| Text WebBrowser | Text | Multi-step web search and page parsing (`LocalGoogleSearchTool`, `VisitTool`). |
| Image Information | Image | Reverse image search and provenance analysis (`GoogleLensSearchTool` with SSIM filtering). |
| Literature Search | Scholarly text | Peer-review retrieval and PDF parsing (Scholar websites, `SpringerDownloadAndParseTool`). |
| File Processing | Documents | Typed file parsing: `PDFTool`, `DOCXTool`, `XLSXTool`, `PPTXTool`. |
| OCR | Image text | Manuscript transcription (Transkribus, Asian-script OCR). |
| Speech Recognition | Audio | Whisper-based transcription with LLM correction. |
| Translator | Multilingual text | Bidirectional translation with provenance preservation. |
| Video | Video | Frame extraction (yt-dlp, OpenCV). |

*Table 11.* Function Overview of HistAgent Specialist Agents.

## C. HistAgent Description

### C.1. Manager Agent

The Manager Agent is the central coordinator. It parses user requests, selects which agent to invoke, gathers their outputs, checks completeness, and synthesizes a final response.

**Functionality.** The Manager Agent parses the user request to identify required modalities, then runs a ReAct-style loop as a Smolagents `CodeAgent`. In each iteration, the LLM emits a Python code snippet that calls sub-agent functions or tools; the snippet executes in a secure sandbox and writes its output to shared memory. The Agent then validates each result (checking exact quotes and bibliographic data) and repeats until a `final_answer(...)` call. At that point, it merges all verified outputs into a single, citation-ready response.

### C.2. Text WebBrowser Agent

The Text WebBrowser Agent handles open-domain web searches and browsing. It simulates a multi-step search and extracts structured content from web pages using a suite of navigation and inspection tools.

**Functionality.** The agent starts by refining the user's query via `LocalGoogleSearchTool` into an optimized search prompt. It can also perform standard queries via `SearchInformationTool`. For each target, it invokes `VisitTool` to load the page, `DownloadTool` to save binary files when needed, and `ArchiveSearchTool` to retrieve historical snapshots. When encountering long pages, it scrolls with `PageUpTool` and `PageDownTool`, and it locates specific terms using `FinderTool` and `FindNextTool`. Non-HTML content, like plain text files, PDFs, or video transcripts, is converted to text through `TextInspectorTool`. All outputs are recorded to shared memory for downstream integration.

### C.3. Image Information Agent

The Image Information Agent focuses on visual inputs. It performs reverse image search and follows up with targeted page visits to uncover context and provenance of the visual input.

**Functionality** The Image Information Agent is selectively invoked when the task involves image content that may offer contextual or evidential value. Upon receiving an image, the Image Information Agent uploads it to a public host and runs a reverse search using `GoogleLensSearchTool`. The system extracts associated links, titles, and descriptions from matched pages to identify how the image is used online, e.g., in auction listings, academic articles, or museum databases. To improve match quality, the agent optionally computes similarity scores (e.g., SSIM) between the original image and search results to highlight high-confidence matches. To gather in-depth metadata, it visits selected pages with `VisitTool_Image`. By grounding image interpretation in its real-world usage context, the agent enables accurate and verifiable historical reasoning over visual inputs.

### C.4. Literature Search Agent

The Literature Search Agent is a specialized module for retrieving and grounding answers in peer-reviewed academic sources. It combines an LLM-driven browser workflow with multiple tool wrappers, covering Google Books, Google Scholar, general web searches, and Springer Nature's API, to locate exact phrases, download accessible PDFs, and extract verbatim quotes along with full citation metadata.

**Functionality.** The agent issues a multi-stage search: it first queries Google Books (with random domain rotation and robots.txt compliance) via `BookMatchExtractorTool` and `DirectGoogleBooksCrawlerTool`, extracting highlighted snippets and page numbers; if needed, it proceeds to Google Scholar using `LiteratureSearchingTool` and `RelevantLiteratureFinderTool` for broader article discovery; it falls back on `GeneralBrowserTool` for additional context. Freely accessible PDFs are downloaded and parsed with `SpringerDownloadAndParseTool` (via LlamaParse) or `SpringerSearchTool`/`SpringerStructuredSearchTool` for structured Springer Nature queries. Throughout, it preserves exact wording, records source URLs, citation counts, publication details, and assembles all findings into a coherent, verifiable response. it can locate sources that support factual claims, provide historical context, or contain exact wording required for exactMatch tasks. It returns full bibliographic metadata, quoted excerpts, and links to original publications, thereby ensuring academic-level faithfulness and verifiability.

### C.5. File Processing Agent

This agent manages non-HTML files—documents, spreadsheets, presentations, and images—by routing them to the appropriate tool.

**Functionality.** When a file is received, the file processing Agent automatically detects its type and selects the corresponding tool: `PDFTool` for PDFs, `DOCXTool` for Word documents, `XLSXTool` for spreadsheets, and `PPTXTool` for presentations. For images that require analysis beyond OCR, it uses `ImageAnalysisTool` to extract charts or figures. All extracted text or structured data is returned in a format that downstream agents or the Manager Agent can incorporate into their reasoning.

### C.6. OCR Agent

The OCR Agent specializes in extracting textual information from images using optical character recognition. It is invoked for screenshots, scanned documents, historical manuscripts, or photos containing embedded text. The agent supports multiple languages and detects the language in the image. Upon invocation, it returns the raw text content detected, enabling HistAgent to convert unstructured visual inputs into machine-readable text for further processing.

**Functionality.** When given an image path, the agent loads and encodes the file and uses an LLM to determine whether the content is best handled by a specialized OCR model for Asian scripts or by a Transkribus-based OCR service for Western-language manuscripts. For Western texts, it publishes the image to a public URL, submits it to the Transkribus engine, waits for a PAGE XML transcription, and extracts the detected lines. For Asian scripts, it sends the image data directly to the dedicated OCR model. The resulting raw transcription is then passed through an LLM prompt that repairs recognition errors and preserves historical or stylistic features. Both the original and refined transcripts are saved to a `.txt` file and returned. If no valid text is extracted, the agent falls back to generating a detailed visual description via the LLM, highlighting any readable text, symbols, or key visual elements.

### C.7. Speech Recognition Agent

The Speech Recognition Agent converts audio files (MP3, WAV, etc.) into text using Whisper for transcription and an LLM for error correction, summary, and key-point extraction, enabling HistAgent to incorporate oral historical sources.

**Functionality.** When given an audio file path, the agent verifies its existence and measures its size; if the file exceeds 25 MB, it divides the recording into equal segments; otherwise, it uses the file as a whole. Each segment is sent to Whisper for transcription, and the concatenated raw transcript is submitted to an LLM prompt that preserves all content while correcting recognition errors and generating an "Optimized Transcription" section, a brief "Summary," and a "Key Points" list. Both original and refined texts are saved to a `.txt` file in the output directory for humans' reference, and a formatted string containing both versions is returned, with any errors caught and reported.

### C.8. Translator Agent

The Translator Agent converts text into a specified target language, including support for both widely spoken and less common languages like Armenian and Sanskrit, and delivers a clear, formatted output showing the original and translated text.

**Functionality.** The Translator Agent handles automatic translation of textual content between multiple languages. It automatically detects the source language and supports both widely used and lower-resource languages. In historical tasks, it is particularly useful for translating foreign-language sources such as Armenian manuscripts, Sanskrit inscriptions, or early regional documents in Latin or Classical Arabic. The translated output allows HistAgent to reason across linguistic boundaries and integrate multilingual content into its historical analysis pipeline.

### C.9. Video Agent

The Video Agent downloads the video from the given link and extracts still frames at a user-specified rate to support visual analysis.

**Functionality.** When given a video URL, the agent downloads the best-quality video with yt-dlp, uses OpenCV to extract frames at the specified rate, saving each as a timestamped JPEG, and writes a summary file containing the video's title, duration, resolution, frame count, and output directory. It then returns a brief report with those key details and file locations.

## D. Literature Search Agent Details

### D.1. Priority Search Protocol

A core challenge in automating academic research is ensuring that retrieved information aligns with scholarly standards. To address this, our system introduces a priority-based retrieval mechanism that favors academically reputable sources over general-purpose content. This design choice reflects the distinct requirements of academic tasks, where the reliability of information is critical. The protocol acts as a persistent control signal for the agent, influencing both the decision to invoke tools and the interpretation of retrieved results.

### D.2. Toolkits for Literature Search

Efficient execution of the academic-first retrieval strategy requires structured tool use. To this end, our system integrates a suite of specialized retrieval tools, each targeting different tiers of source quality. All source queries, are first conducted via the `LiteratureSearchingTool` to search from scholarly databases, and then to `GeneralBrowserTool` interface for general search. The queries will also use more specialized APIs when needed. For example, Springer-specific queries are handled using the `SpringerSearchTool`, and focused access to Google Books is enabled through the `DirectGoogleBooksCrawlerTool`. Most of the retrieved documents can be extracted directly, while pdfs could be parsed via the `Llama-Parse` API for structured processing. This modular architecture ensures that each retrieval action is consistent with the source prioritization strategy established earlier.

### D.3. Human-controllable Agent Customization

Autonomous retrieval must adapt to varying task demands and resource limitations. Our system provides users with explicit control over agent behavior through a small set of interpretable parameters. These include limits on reasoning steps and re-planning intervals, which allow users to trade off between search depth and computational cost. Such configurability is critical for real-world deployment, where users may need to enforce practical constraints while preserving the academic quality of the output.

### D.4. Clearly labeled Sources

Trust and traceability are essential in academic applications. To meet these needs, our system ensures that all retrieved content is accompanied by clearly labeled source metadata. This includes bibliographic information, retrieval URLs, and quoted evidence, all linked to the reasoning trace. These annotations allow users and downstream systems to verify claims, reproduce retrieval steps, and integrate results into formal academic workflows.

# E. Limitations

Our HistBench has two limitations. First, all the questions are closed questions with an only exact answer to simplify the evaluation criteria, which lead to a monotony in the types of questions. In fact, handling of questions with an open answer is a critical part of historical research. The evaluation of questions with an open answer needs an intricate and dynamic evaluation criterion. The determination of this evaluation criterion is difficult and requires further exploration. Second, although contributors are required to formulate questions with objective answers, the design of questions is still inevitably influenced by the contributors' cognition and the current research limitations. For example, due to differences in material selection and interpretation,the academic world may have different answers to an objective question, while the contributors may choose the information they have accessed and believe to provide the answers for the questions. This appears to be an inevitable drawback.

# F. Broader Impacts

**Positive Impacts.**   This work enables more accurate and scalable analysis of historical materials, potentially benefiting educators, researchers, and cultural institutions. HistAgent could help democratize access to historical knowledge across linguistic and archival boundaries, support the preservation of underrepresented histories, and promote computational methods in the humanities. By improving AI's interpretive capabilities, it may also foster interdisciplinary collaboration between computer science and historical research.

**Negative Impacts.**   HistAgent's outputs, if misused or misunderstood, could lead to the spread of misleading historical narratives, especially when dealing with ambiguous or contested sources. There is also a risk of over-reliance on AI systems in academic workflows, potentially marginalizing expert judgment. Furthermore, the use of OCR and translation tools on sensitive or culturally specific materials raises ethical concerns regarding misinterpretation, bias amplification, or the unintended exposure of vulnerable archives.

# G. Declaration of LLM Usage

This research involves the use of large language models (LLMs) as core components of the experimental framework. Specifically, the HistAgent (our method) and ODR-smolagents (baseline) both rely on GPT-4o as the underlying language model. The purpose of our experiments is to isolate and compare the performance impact of different agent architectures, with the LLM held constant across systems to ensure a fair comparison. The differences in outcome are therefore attributed to the design of the agent frameworks, not the underlying model capabilities.

Each agent operates in a fresh session per question. We impose a fixed budget on the number of tool invocations (agent calls) to avoid infinite loops and normalize the computational cost across runs. The output of each agent is a final answer that we compare against ground-truth responses.

In addition to GPT-4o-based systems, we also benchmark against DeepSeek-R1, GPT-4o, o4-mini-high, o3, and Grok 3 on HistBench. These models are used in their native agentic or search-augmented setups, consistent with how they are deployed in real-world scenarios. All these models have online search capabilities and demonstrate strong reasoning abilities in open-domain tasks.

The LLMs are strictly part of the technical comparison and evaluation of agent performance, which forms the core contribution of the work.

# H. Details in Experiment

## H.1. Experiment Compute Resources

We use M1 chip with 16G memory for all of the experiments. While our histagent and the ODR-smolagents requires about 512 MB for running one question.

For the time consumption, our histagent needs about 20 minutes per question. The ODR-smolagents needs about 5 minutes per question. The GPT-4o with web search needs about 2 minutes per question. The other baselines are run manually.

## H.2. Experiment Setup and Datasets

**HistBench**. HistBench is a history-specific benchmark we construct, containing 414 history questions categorized into three difficulty levels (Level 1–3).

**GAIA.** To test the generalization of our framework, we also test our HistAgent on the validation set of GAIA benchmark(Mialon et al., 2023), which includes 165 tasks.

**HLE History Subset.** We consider the HLE history subset of 56 questions from the Humanity's Last Exam benchmark(Phan et al., 2025).

We compare our HistAgent against open Deep Research (ODR-smolagents)(Roucher et al., 2025), an open-source reproduction of OpenAI's Deep Research agent by HuggingFace/smolagents. It uses a Manager agent and a Text Web Browser Agent with a visualizer tool and a file processing tool. Both systems are run under identical environments for fairness. We use the same underlying language model (GPT-4o) for both HistAgent and ODR-smolagents, ensuring that differences in performance stem from the agent architecture rather than the base model. Each question is posed to the agent in a fresh session. We limit the total number of tool invocations (agent calls) per question to a fixed budget to prevent infinite loops and to enforce a fair comparison. Both systems output a final answer for each question, which we compare to the ground-truth answer. Additionally, we compare our results with DeepSeek-R1 (Guo et al., 2025), GPT-4o (Hurst et al., 2024), o4-mini-high (OpenAI, 2025), o3 (OpenAI, 2025), and Grok 3 (xAI, 2025) on HistBench for a thorough comparison. All these LLMs are equipped with online search capabilities, enabling them to be very strong baselines.

## H.3. Detailed Evaluation Metrics

**Metric for HistBench.** For the HistBench benchmark, we measure accuracy as the percentage of the 414 questions for which the agent's response is judged correct by both the LLM judge and professtional validation. The evaluation proceeds in three steps:

- **Structured Output:** Each response is composed of a concise final answer and a structured reasoning summary (shown in Appendix H.6) that logs tools used, information sources, and step-by-step logic.

- **LLM Judging:** We run LLM as a judge to issue a binary judgment ("Correct"/"Incorrect") based on semantic equivalence, completeness of key facts, and logical coherence. The prompt template is an adaptation of the evaluation prompt of HLE, which is shown in Appendix H.5.

- **Human Expert Validation:** To validate the LLM's judgment quality, we randomly sample 100 examples across Level 1, Level 2, and Level 3 tasks. Each sample includes both the final answer and its reasoning summary. These are reviewed by the original authors of the questions, who assess whether the correct answer could have been obtained merely by coincidence, verify the reliability and factual accuracy of every cited source, and confirm that the sequence of reasoning steps genuinely supports the final answer. An answer is labeled "Correct" only if it satisfies all of these criteria; otherwise, it is marked "Incorrect". This human expert evaluation not only provides an external check on model-assigned labels but also reveals edge cases and ambiguities in question phrasing or evaluation criteria, allowing us to iteratively refine and improve the overall quality and consistency of our HistBench.

We define accuracy on HistBench as the percentage of the 414 questions for which the response is both "Correct" by the LLM judge and "Correct" by the author (100 samples selected). We report this metric overall and separately for Level 1, Level 2, and Level 3.

**Metric for GAIA.** We employ the official scoring function published on Hugging Face. Each GAIA question has a standard answer key; the scoring function handles normalization (e.g., case, punctuation) and computes an exact-match score. We report the overall GAIA accuracy as defined by that function.

**Metric for HLE History Subset.** For the 56 expert-level questions in the HLE History Subset (curated from Humanity's Last Exam), evaluation mirrors the initial stages of the main HLE benchmark protocol. Each response must first provide a Structured Output, comprising a concise final answer and a detailed reasoning summary (as described for the HLE benchmark before). Subsequently, we employ LLM-based Judging, where an LLM assesses the response for semantic equivalence to the ground truth, factual completeness, and logical coherence, issuing a binary judgment ("Correct"/"Incorrect") as the

HLE official scoring function released on github. Accuracy for this subset is then calculated as the percentage of questions deemed "Correct" by the LLM judge.

## H.4. Analysis for Performance on GAIA

We evaluate our HistAgent on the 165-question validation subset, using the accuracy as the main metric. The split includes 53 Level 1, 86 Level 2, and 26 Level 3 questions, covering open-book fact finding, tool use, and multimodal reasoning. Our HistAgent, based on the Claude-3.7-sonnet, answers 99 questions correctly and reaches an overall accuracy of 60.00% pass@1. The baseline, open Deep Research by HuggingFace/smolagents, records 55.15% on the same split. These results show that HistAgent, as a domain-specific agent, can generalise reliably beyond its original scope. More details are provided in Table 12, which presents the complete level-wise breakdown.

| Agent | Model | Average | Level 1 | Level 2 | Level 3 |
|---|---|---|---|---|---|
| HistAgent | Claude–3.7–sonnet | **60.00** | 69.81 | 61.63 | 34.62 |
| ODR-smolagents | o1 | 55.15 | 67.92 | 53.49 | 34.62 |

*Table 12.* Performance accuracy (%) on the GAIA validation set (pass@1)

## H.5. Prompt Template

### JUDGE_PROMPT Template

```
JUDGE_PROMPT = """You are a fair evaluator.  Judge whether the
following [response] to [question] is semantically consistent with the
[correct_answer] below.
[question]:  {question}
[response]:  {response}
[correct_answer]:  {correct_answer}
When you judge, consider only whether the core meaning and all necessary key
points in the response match the correct answer.  Even if wording or format
differs, treat equivalent semantics as correct.  Treat missing key points
or any substantive error or omission as incorrect.  For numerical answers,
a small rounding difference is acceptable.  Tolerate substantive deviations
from the correct answer.  If the extracted_final_answer is a more specific
instance of the correct_answer (for example, "Pieter Schenk II" vs "Pieter
Schenk"), and it still contains the core string of the correct_answer, treat
it as correct.
Please output exactly in the format and criteria specified below:
extracted_final_answer:  The final exact answer extracted from the
[response].  Put the extracted answer as 'None' if there is no exact, final
answer to extract from the response.
reasoning:  Explain why the extracted_final_answer is correct or incorrect
based on [correct_answer], focusing only on if there are meaningful
differences between [correct_answer] and the extracted_final_answer.  Do
not comment on any background to the problem, do not attempt to solve the
problem, do not argue for any answer different than [correct_answer], focus
only on whether the answers match.
correct:  Answer 'yes' if extracted_final_answer matches the
[correct_answer] given above, or is within a small margin of error
for numerical problems.  Answer 'no' otherwise, i.e.  if there is any
inconsistency, ambiguity, non-equivalency, or if the extracted answer is
incorrect.
```

```
confidence:  The extracted confidence score between 0% and 100% from
[response].  Put 100 if there is no confidence score available."""
```

## H.6. Summary Case Study

---

**Summary Case Study**: Summary for Level 2_30

**Question ID:** level_2_30
**Question:** What is the name of the man in this picture?
**Answer Type:** exactMatch
**Data Requirement:** level_2_30.png
**Data Type:** file
**Our Answer:** Veit Rudolph Speckle
**Correct Answer:** Veit Rudolph Speckle / Veit Rudolf Specklin
**Is Correct:** Yes
**File:** Historical/Historical/level_2_30.png
**Model:** gpt-4o

**Summary Report**
**1. Tools Used and How They Were Used:**
- **Image Analysis and Reverse Image Search**:
- Purpose: To identify the individual in the image "Historical/Historical/level_2_30.png".
- Method: Used the `image_information_agent` to perform a reverse image search to identify the name attached to the image and gather related historical context.
- Result: The reverse search linked the image to Veit Rudolph Speckle, associated with botanical illustrations for Leonhart Fuchs' "De Historia Stirpium".
- **Literature Search Agent**:
- Purpose: To find scholarly literature verifying the exact match for Veit Rudolph Speckle's association with Fuchs' work.
- Method: Used the query "Veit Rudolph Speckle is one of the engravers associated with Leonhart Fuchs' 'De Historia Stirpium.'".
- Result: Confirmed findings that Speckle was a renowned engraver for this botanical work, highlighted as responsible for key woodcut illustrations.
- **Web Search**:
- Purpose: To access additional context and verify the scholarly data.
- Method: Conducted a search on Google and accessed Google Books, Google Scholar, and other scholarly sources.
- Result: Successfully retrieved confirmation and additional scholarly and book references regarding the engravers and contributors to Fuchs' herbal.
**2. Detailed Information Sources:**
- **Wikidata**
- URL: Wikidata: Veit Rudolph Speckle
- Quote: "The primary identification linked it to Veit Rudolph Speckle, as noted on Wikidata..."
- Credibility: Open user-contributed database but verifies common scholarly facts.
- **Article in "The World of Plants in Renaissance Tuscany" by Cristina Bellorini**:
- Quote: "Veit Rudolph Speckle was an engraver for Leonhart Fuchs' 'De Historia Stirpium'."
- Credibility: Academic book providing historical insight into botanical studies during the Renaissance.
- **Google Scholar Sources**:
- Quotes: "Veit Rudolph Speckle was responsible for the woodcut engravings in Leonhart Fuchs' herbal." and "Recognized as 'by far the best engraver in Strasbourg...".
- URLs: Retrieved through exploratory searches within Google Scholar.
- Credibility: Google's academic resource is known for aggregating reputable, peer-reviewed materials.
**3. Reasoning Process and Logic Steps:**

---

- **Identification**:
- Initial determination of the man in the image was made through reverse image search, identifying him as Veit Rudolph Speckle.
- **Verification**:
- Utilized literature searches on academic databases (Google Scholar, books) to verify Speckle's role in creating illustrations for Fuchs' botanical book.
- **Cross-verification**:
- Multiple sources including books and academic papers were consulted to confirm Speckle's work as an engraver for Leonhart Fuchs' book "De Historia Stirpium."
- **Exclusion of Other Answers**:
- The reverse image search did not present any alternative credible identity, leading to a focused inquiry on Speckle which was consistently supported by scholarly resources.
**4. Answer Quality and Reliability Analysis:**
- **Reliability**: High
- Given the cross-corroboration from reliable academic texts and reputable databases (Google Scholar, academic books), the reliability is high.
- **Assumptions, Weaknesses, Uncertainties**:
- Assumptions largely relied on the historical accuracy maintained by sources. Lack of web search limits alternative verifications.
- **Sufficiency and Consistency**:
- The evidence gathered was sufficient, consistent, and convergent from independent, reliable sources, affirming the credibility of the information.
- **Suggestions for Improvement**:
- Include a broader web search to capture any contemporary assessments or potential misattributions regarding this artwork or engraver. Suggested keywords: "Veit Rudolph Speckle", "Leonhart Fuchs botanical engravings".
- **Web Search Observation**:
- Although literature and specific academic sources provided strong backing, the integration of broader web search could improve the reliability by encompassing wider perspectives or additional public domain resources.

