# OpenReview forum: "On Path to Multimodal Historical Reasoning: HistBench and HistAgent"
_ICML.cc/2026/Conference — ICML 2026 regular_

### Official Review · Reviewer_6Nsc · 2026-03-10

**Soundness:** 3
**Presentation:** 3
**Significance:** 2
**Originality:** 3
**Overall Recommendation:** 4
**Confidence:** 3

**Summary:**

This paper studies multimodal historical reasoning for large language models and agents. The authors introduce HistBench, an expert-curated benchmark spanning multiple modalities, languages, and difficulty levels, and also propose HistAgent, a domain-specific agent for processing diverse historical sources. The paper reports that HistAgent outperforms several strong general-purpose baselines on HistBench while preserving competitive performance on GAIA.

**Compliance With Llm Reviewing Policy:**

Affirmed.

**Final Justification:**

The rebuttal addressed my concerns, so I maintain my score unchanged.

**Key Questions For Authors:**

1. Why is human verification performed on only a limited sample, and what evidence shows that the LLM judge is stable across languages, modalities, and difficulty levels?
2. How much of HistAgent's gain remains if one controls more carefully for access to retrieval tools and curated workflows?

**Limitations:**

The paper should discuss more explicitly the subjectivity of evaluating open-ended historical reasoning, the limits of benchmark coverage, and the dependence of HistAgent on curated retrieval and workflow design.

**Strengths And Weaknesses:**

Strengths:

The topic is interesting and clearly underexplored. The benchmark design also appears thoughtful, especially in its multilingual and multimodal scope.

Weaknesses:

1. For an open-ended, multilingual, multimodal benchmark of historical reasoning, the paper relies too heavily on LLM-based judging.

2. HistAgent is interesting, but it is not clear how much of its gain comes from a principled general agent design versus benchmark-coupled workflow engineering and access to curated retrieval tools.

3. The dataset size is insufficient. It is limited for a task with such high variance and interpretive ambiguity.

---

> ### Author Rebuttal · Authors · 2026-03-31
>
> > **Reviewer's comment**: For an open-ended... // Why is human verification ...
>
> **Response 1**: We thank the reviewer for this important point. We agree that evaluation in open-ended, multilingual, and multimodal settings should be clearly justified. In our benchmark, most questions are multiple-choice or exact-match, where LLM-as-a-judge mainly handles normalization (e.g., formatting and cross-lingual equivalence), introducing limited variance.
>
> LLM-based evaluation is widely adopted in prior work [1–2] and has been shown to align well with human preference [3]. Our protocol follows this paradigm for consistency and scalability. Moreover, recent multilingual benchmarks [1, 4, 5] also adopt LLM-as-a-judge, supporting its applicability in cross-lingual settings. Low-resource and historical language cases in our benchmark are designed to minimize ambiguity (e.g., multiple-choice or multilingual exact-match with English references), reducing reliance on the judge’s language fluency.
>
> To further ensure reliability, we conduct human expert validation on a sampled subset of 100 questions. Given the high cost of large-scale human evaluation, this validation serves to assess both HistAgent performance and the consistency of LLM-based judgments with human evaluation. The agreement analysis shows that LLM judgments align well with expert annotations.
>
> > **Reviewer's comment**: HistAgent is interesting... // How much of HistAgent's...
>
> **Response 2**: We thank the reviewer for this important point. To isolate the effect of agent design, we compare under the same backbone (GPT-4o) against ODR-smolagents, ensuring gains are not due to model scaling. Moreover, HistAgent’s components (e.g., OCR, translation reflect general capabilities for historical reasoning rather than benchmark-specific heuristics, and consistent gains on HLE-History and GAIA suggest generalization.
> We further conduct a new partial ablation on a 30-question subset due to cost constraints:
> | Metric        | OCR + Web Search | File Tool + Web Search | Translator + Web Search | Web Search Only (ODR) |
> |---------------|------------------|------------------------|-------------------------|------------------------|
> | Accuracy (%)  | 31.0| 25.0| 23.3| 20.0  |
>
> These results show that each component contributes to performance, with OCR providing the largest gain, supporting that improvements arise from the agent design rather than benchmark-specific engineering.
>
> > **Reviewer's comment**: The dataset size...
>
> **Response 3**: We thank the reviewer for this helpful suggestion. We agree that dataset scale is important for high-variance tasks. HistBench is designed as a high-quality evaluation benchmark, prioritizing rigor and discriminative difficulty over scale.
> The initial pool contains **1034 candidate questions**, which are refined to 414 through multi-stage screening to ensure rigor and discriminative difficulty.
>
> We note that the effort involved in constructing such data: the time required per question varies with difficulty (e.g., Level 1 questions are relatively faster, while higher-level questions require substantially more time for sourcing, verification, and interpretation), with an average effort on the order of tens of minutes per question, often involving iterative revisions. In addition, evaluation with agent pipelines is computationally intensive (approximately \$4 per question), which constrains scaling.
>
> Despite its size, HistBench shows strong discriminative power across models and difficulty levels, indicating that carefully curated items provide meaningful evaluation signals. We compare with representative benchmarks:
> | Benchmark | # Questions | Domain  |
> |--------------------------|------------|--------------------------------|
> | GAIA (validation)        | 466        | General reasoning              |
> | SWE-bench Verified [6]      | 500        | Software engineering           |
> | MiniF2F [7]  | 488        | Formal mathematics             |
> | **HistBench (ours)**     | **414**    | Multimodal historical reasoning |
>
> > **Reviewer's comment**:  Limitations: The paper should discuss...
>
> **Response 4**: We thank the reviewer. These aspects are already discussed in Appendix E, including the use of closed-form questions for evaluation and the inherent subjectivity in historical interpretation. We agree that this should be more explicit in the main text and will update it in the revision.
>
> [1] Phan, Long, et al. "Humanity's last exam."
> [2] Bai, Ge, et al. "Mt-bench-101"
> [3] Zheng, Lianmin, et al. "Judging llm-as-a-judge with mt-bench and chatbot arena."
> [4] Du, Yexing, et al. "Ccfqa"
> [5] Huang, Xu, et al. "Benchmax"
> [6] Jimenez, Carlos E., et al. "Swe-bench"
> [7] Zheng, Kunhao, et al. "Minif2f"

---

> > ### Author Rebuttal · Reviewer_6Nsc · 2026-04-03
> >
> > The rebuttal addressed my concerns, so I maintain my score unchanged.

---

### Official Review · Reviewer_mCfs · 2026-03-11

**Soundness:** 2
**Presentation:** 3
**Significance:** 2
**Originality:** 3
**Overall Recommendation:** 3
**Confidence:** 3

**Summary:**

The submission makes two main contributions. HistBench is a question-answering dataset that includes multimodal documents in 29 languages, spanning different regions, historical periods, and topics. The dataset curation process is supported by domain experts with different degrees of expertise. HistAgent is a specialized agent designed to answer heterogeneous questions such as those included in HistBench. The evaluation compares HistAgent with a generic agent, base models, and reasoning models on HistBench and history-related subsets of other Q&A datasets.

**Compliance With Llm Reviewing Policy:**

Affirmed.

**Final Justification:**

I thank the authors for the supplementary clarifications following the acknowledgment.
Considering the strengths and weaknesses of the submission, I maintain the original score.

**Key Questions For Authors:**

-Q1: Given the ambition to cover different domain-related dimensions (languages, regions, periods, topics) and the participation of 40 experts, why weren’t more questions included in the curated dataset?

-Q2: Why weren’t o3 and 04-mini tested on the GAIA and HLE subsets and integrated as backbone models in HistAgent? Such integration would have enable a clearer understanding of the added value of the specialized components of the agentic architecture.

-Q3: Why wasn’t an ablation study included in the paper? It would have provided insight into the role of HistAgent’s core components.

-Q4: Similarly, why doesn’t the submission discuss results per language, region, historical period, or type of input? Such granular analysis would better highlight its added value compared with the baselines.

**Limitations:**

The limitations and impact sections are deferred to Appendices E and F.

**Strengths And Weaknesses:**

Strengths

-S1: The submission addresses a relatively overlooked domain and proposes a curation pipeline that relies on strong domain expertise. The inclusion of multimodal documents, of several languages, and of topics covering different world regions is appreciated.

-S2: The proposed dataset curation is appropriate. The strong reliance on expertise improves the domain relevance of the questions. The stratification per difficulty level uses relevant criteria. The filtering process, based on LLM and human inputs, excludes trivial questions and those considered unfit by experts.

-S3: The results indicate that HistAgent effectively improves results when compared with the backbone model upon which it relies. It is also competitive with stronger models on easy questions.

Weaknesses

-W1: While the dataset curation effort is worthy, the dataset size is limited given its ambition to cover different languages, regions, and historical periods. With 40 participants, the effort needed to scale the dataset up 10x should not be prohibitive. A larger dataset would allow a more granular evaluation across the different dimensions mentioned above.

-W2: The related work should be included in the main text, not in an appendix. As per ICML reviewer instructions, “Reviewers are encouraged (but not required) to read any Supplementary Material provided alongside their assigned submissions.”.  The submission could cite KM-MHISTO and evaluate HistAgent on it: https://aclanthology.org/2025.findings-acl.1042.pdf
This dataset focuses on long-tail historical questions that are difficult for LLMs to answer. Testing HistAgent on it would increase the robustness of the evaluation.

-W3: The result reporting is partly misleading. HistAgent does not outperform o3 on Level 2 and Level 3 questions. A finer-grained analysis would be needed in this case.  Also, o3 and o4-mini should be tested on HLE and GAIA to have a better understanding of baseline behavior.

-W4: Why wasn’t the HistAgent paired with stronger foundation models? Those cited L419-L422 are available for quite some time now. This would enable the reader to understand the added value of the agentic architecture.

-W5: The submission lacks a proper ablation study that would enable the understanding of the different components’ roles. Such a study should at least remove the core components, such as the Literature Search, the OCR, or the Translator.

-W6: The analysis would be much more insightful if done per language, document type, region, period, etc. To remain relevant and robust, such analysis would require a much larger and more balanced dataset.

-W7: The evaluation should include a complexity evaluation to highlight the compromise between effectiveness and efficiency. How long does it cost to run HistAgent versus GPT-4o versus o3 and o4?

Minor

-M1: Figure 2c does not support the claim of geographic diversity well. The overview should include the number of questions per major region of the world. For instance, how many Africa-related questions does HistBench include?

-M2: The dataset curation process should be described more rigorously. How many participants per expertise level? What are the demographics of the participants? How many questions per difficulty level are there in the final dataset?

-M3: Is the automated access to resources such as Google Scholar or Google Lens (Table 2) legal? If so, on what ground?

---

> ### Author Rebuttal · Authors · 2026-03-31
>
> > **Reviewer's comment**: W1: While the dataset.. // Q1: Given the ..
>
> **Response**: We thank the reviewer for this helpful suggestion. Due to space constraints, please refer to Response 3 in our rebuttal to Reviewer 6Nsc.
>
> > **Reviewer's comment**: W2: The related work..
>
> **Response**: We thank the reviewer for this helpful suggestion and will cite KM-MHISTO in the revised version. Regarding KM-MHISTO, we appreciate the recommendation. We conduct a new preliminary evaluation for HistAgent and GPT-4o on 50 randomly sampled questions from the dataset; due to the cost of approximately $2.5 per question in this datatset, large-scale evaluation is currently not feasible. HistAgent (GPT-4o) achieves 0.90 pass@1, compared to 0.50 pass@1 for GPT-4o, indicating strong performance on long-tail historical questions. These results further support HistAgent’s generalization beyond HistBench.
>
> > **Reviewer's comment**: W3: The result... // Q2: Why weren’t ...
>
>
> **Response**:
> We thank the reviewer for this important point and insightful suggestion. In particular, performance varies across difficulty levels, with stronger reasoning models (e.g., o3) showing advantages on higher-level questions that require more intensive reasoning, while HistAgent uses GPT-4o.
> We also agree that evaluating stronger models  would further enrich the baseline analysis. Our current submission focuses on matched-backbone comparisons (e.g., with ODR-smolagents under GPT-4o) to isolate agent design effects.
> Due to budget limit, we conduct new experiments with **o4-mini-high on the HLE history subset**, obtaining 4/56 (7.1%) pass@1, which is much lower than HistAgent. For reference, recent reported results on whole HLE show that o3-mini achieves ~10.5% (balanced) to 13.0% (high).
>
>
> > **Reviewer's comment**: W4: Why wasn’t ..// Q2: and integrated ...
>
> **Response**: We thank the reviewer for this suggestion. Using stronger models significantly increases cost (already about $4 per question for GPT-4o). Our goal is to isolate the effect of the agent design, so we focus on controlled comparisons under the same backbone (GPT-4o with ODR). We agree that stronger models are worth evaluating and will include such results where feasible in the revision.
>
>
> > **Reviewer's comment**: W5: The submission ... // Q3: Why wasn’t ...
>
> **Response**: We thank the reviewer for this helpful suggestion. Due to space constraints, please refer to Response 2 in our rebuttal to Reviewer 6Nsc.
>
> > **Reviewer's comment**: W6: The analysis... // Q4: Similarly, why...
>
> **Response**: We thank the reviewer for this valuable suggestion. We include a modality-based breakdown showing that while strong LLMs with search remain competitive on text-only questions, HistAgent achieves consistent gains on multimodal questions, outperforming all baselines across difficulty levels, which indicatesimprovements come from processing non-textual historical evidence. Due to space limits, please refer to the [reference link](https://anonymous.4open.science/r/icml_hist-B665/) for the table.
>
>
> > **Reviewer's comment**: W7: The evaluation...
>
> **Response**: We thank the reviewer for this helpful suggestion. Following this, we conduct a new preliminary runtime comparison on a representative example: **GPT-4o (8.4s), o3 (41.2s), o4-mini-high (51.4s), and HistAgent (257s)**. While full-scale measurement is computationally expensive, this result provides a useful reference for the efficiency–performance trade-off.
>
> > **Reviewer's comment**: M1: Figure 2c...
>
> **Response**: We thank the reviewer for this helpful suggestion. We provide an explicit geographic breakdown of HistBench. The dataset covers a broad range of regions, including East Asia (186, 44.9%), Europe incl. Russia (147, 35.5%), North America (31, 7.5%), Middle East & Central Asia (14, 3.4%), South Asia (7, 1.7%), Africa (7, 1.7%), Southeast Asia (5, 1.2%), Oceania (5, 1.2%), and Latin America & Caribbean (4, 1.0%), with a total of 414 questions. For example, HistBench includes 7 Africa-related questions, as specifically raised by the reviewer.
>
> > **Reviewer's comment**: M2: The dataset...
>
> **Response**: Contributors span three expertise levels with approximately 15 (Level 1), 48 (Level 2), and 41 (Level 3) participants, with partial overlap across levels. We did not collect detailed demographic attributes to preserve privacy; contributors are primarily students, graduate researchers with relevant knowledge, and domain experts. The final dataset contains 414 questions: Level 1: 166 (40.1%), Level 2: 172 (41.5%), Level 3: 76 (18.4%).
>
> > **Reviewer's comment**: M3: Is the automated...
>
> **Response**: We thank the reviewer for the question. All external access (e.g., Google Scholar, Google Lens) is conducted via SerpAPI, a licensed API that provides compliant access to Google services, rather than direct scraping. This follows standard API usage and academic research practices.

---

> > ### Author Rebuttal · Reviewer_mCfs · 2026-04-01
> >
> > I thank the authors for their effort in addressing the weaknesses.
> > In particular, the new experiments provide clarifications regarding the advantages and limitations of HistAgent.
> > W1/Q1, regarding the dataset's size relative to its stated ambition to cover different periods, regions, and languages, remains unresolved. This negatively affects the thoroughness of the results analysis (W5, WP6). The additional experiment in response to WP6 shows that HistAgent performs better on multimodal questions and worse on text questions.
> > The efficiency analysis (W7) shows that HistAgent is much slower than the baselines, which is a problem in real-world usage.
> > After reading the answers, the other reviews, and the corresponding replies, I maintain the original score.

---

> > > ### Author Response · Authors · 2026-04-05
> > >
> > > # Response to Reviewer mCfs
> > > We sincerely thank the reviewer for the careful follow-up and for engaging with the additional analyses in the rebuttal. We would like to explain the remaining points briefly.
> > > ## 1. Dataset Scale (W1/Q1)
> > > We thank the reviewer for this helpful suggestion. In the revision, we state that HistBench is intended as a **high-quality, expert-curated benchmark for historical evaluation**, and that its scale is consistent with many widely used/accepted benchmarks designed for challenging and diverse evaluation settings.
> > >
> > > | Benchmark | Total # Questions | Languages / Regions | Domain | Venue |
> > > |---|---|---|---|---|
> > > | GPQA [1] | 448 | English only | Graduate-level science | ICLR 2024 |
> > > | InfiBench [2] | 234 | 15 programming languages | Code QA | NeurIPS 2024 |
> > > | Belebele [3]| 900 | 122 languages | Reading comprehension | ACL 2024 |
> > > | GAIA2 [4] | 800 | English only | General agent evaluation | ICLR 2026 |
> > > | GAIA [5] | 466 | English only | General AI assistant | ICLR 2024 |
> > > | MiniF2F [6] | 488 | English only | Formal mathematics | ICLR 2022 |
> > > | SWE-bench Verified [7] | 500 | English only | Software engineering | ICLR 2024 |
> > > | **HistBench (Ours)** | **414** | **29 languages, 20+ regions** | **Multimodal historical reasoning** | — |
> > >
> > > As shown above, many established benchmarks contain fewer than 1,000 questions, and several focus on English-only, single-domain evaluation. In contrast, multilingual and multi-region coverage is intrinsic to historical research, as authentic historical evidence originates from diverse languages, scripts, regions, and modalities. This also makes large-scale expansion more resource-intensive, as expert knowledge is required across different linguistic and cultural contexts.
> > >
> > > HistBench is obtained through a multi-stage filtering process (1,034 → 720 → 414), with expert drafting, verification, and revision for each retained item. We agree that further scaling would enable more detailed per-language and per-region analysis. At the current scale, however, HistBench already supports the main analyses in the paper, particularly along the **difficulty** and **modality** dimensions.
> > >
> > >
> > > ## 2. Efficiency (W7)
> > > We agree that HistAgent is slower than single-model baselines. This is consistent with prior empirical findings in agent benchmarks, where improved task performance comes with increased cost and latency.
> > >
> > > For example, TheAgentCompany [8] benchmark shows that agent-based systems require substantially more steps and incur non-trivial cost (e.g., ≈$4 per task) on real-world tasks. Similarly, SWE-Effi [9] reports that a SWE-Agent with GPT-4o-mini consumes 8.8M tokens and 658s on failed attempts, compared to 1.8M tokens and 167s on successful ones, indicating a ~4× difference in compute usage. Across tasks, agent inference time ranges from tens to hundreds of seconds (e.g., 4.6s to 470.7s per instance).
> > >
> > > Following this line of work, HistAgent prioritizes correctness on complex, multimodal historical reasoning tasks. The additional latency arises from multi-step reasoning and tool use (e.g., OCR, retrieval, translation), which are necessary for reliable evidence grounding in this setting. Empirically, HistAgent achieves 27.54% pass@1 with a GPT-4o backbone, compared to 18.60% for GPT-4o alone (a 48% relative improvement). Since our target use case is low-frequency, high-value historical research rather than high-throughput interaction, we believe this is an acceptable tradeoff.
> > >
> > > Historical reasoning is inherently depth-oriented, where scholars routinely spend hours on a single question. In this context, HistAgent provides a practical tradeoff by substantially improving accuracy while maintaining tractable inference time.
> > >
> > >
> > > ## 3. Modality-Based Analysis (W6)
> > >
> > > We would also like to state that, under matched GPT-4o comparisons, HistAgent does **not** underperform on text-only questions:
> > >
> > > | Subset | HistAgent (GPT-4o) | ODR-smolagents (GPT-4o) | GPT-4o + Search |
> > > |---|---|---|---|
> > > | Text-only Overall | **21.05** | 19.40 | 20.90 |
> > > | Multimodal Overall | **29.44** | 16.08 | 11.19 |
> > >
> > > Under matched GPT-4o comparisons, HistAgent is slightly better on text-only questions and substantially better on multimodal questions. On text-only questions, HistAgent outperforms both ODR-smolagents and GPT-4o + search under the same backbone. Moreover, the improvements are substantial on multimodal questions, where HistAgent significantly outperforms both baselines. This aligns with the design of HistAgent, which focuses on processing non-textual historical evidence (e.g., manuscripts, inscriptions, and images) through OCR, translation, and retrieval tools.
> > >
> > > [1] Rein, David, et al. "Gpqa"
> > > [2] Li, Linyi, et al. "Infibench"
> > > [3] Bandarkar, Lucas, et al. "The belebele benchmark"
> > > [4] Froger, Romain, et al. "Gaia2"
> > > [5] Mialon, Grégoire, et al. "Gaia"
> > > [6] Zheng, Kunhao, et al. "Minif2f"
> > > [7] Jimenez, Carlos E., et al. "Swe-bench"
> > > [8] Xu, Frank F., et al. "Theagentcompany"
> > > [9] Fan, Zhiyu, et al. "Swe-effi:

---

### Official Review · Reviewer_7eNk · 2026-03-11

**Soundness:** 3
**Presentation:** 3
**Significance:** 4
**Originality:** 3
**Overall Recommendation:** 5
**Confidence:** 4

**Summary:**

This paper addresses a notable gap in LLM evaluation: the humanities, and history specifically, have been largely ignored despite being genuinely challenging for AI systems. The authors make two contributions.
The first is HistBench, a 414-question benchmark built from authentic historical materials by over 40 contributors, including professional historians. Questions span 29 languages, 5 modalities, and 3 difficulty levels defined by historian-facing criteria rather than model performance. The benchmark is designed to test the kinds of tasks historians actually do: reading manuscripts, translating dead languages, interpreting visual artifacts, and cross-referencing archival sources.
The second is HistAgent, a domain-specialized agent that pairs GPT-4o with a suite of history-specific tools: manuscript OCR, multilingual translation, scholarly literature search, reverse image search, and audio/video processing. The agent follows a manager-specialist architecture where a central coordinator delegates to specialized sub-agents and iteratively validates evidence before producing a cited response.

**Compliance With Llm Reviewing Policy:**

Affirmed.

**Final Justification:**

My initial concerns centered on three points: the unexplained non-monotonic level-wise performance pattern, the uncontrolled GAIA comparison, and the reliability of LLM judges across low-resource languages. The rebuttal addressed all three satisfactorily. The explanation for the level-wise pattern that LLM pretraining coverage doesn't map onto historian-defined difficulty is well-reasoned and supported by concrete examples. The authors also ran a matched-model GAIA experiment that directly supports the generalization claim. The clarification on judge reliability for low-resource languages, while not exhaustive, is reasonable given that most such questions use deterministic evaluation formats.
The remaining limitations, closed-form question constraint, and the modest novelty of the agent architecture are acknowledged by the authors and are inherent to the current evaluation paradigm rather than failures of execution. The significance of the benchmark contribution is particularly high: this work opens a largely unexplored direction and provides infrastructure that the community can build on.
Taking the rebuttal into account, I am upgrading my recommendation to accept.

**Key Questions For Authors:**

1. Why does performance dip at Level 2 and recover at Level 3? This pattern is unexpected, and the paper doesn't discuss it.
2. The GAIA comparison uses different underlying models for HistAgent and ODR-smolagents. Can you rerun this with matched models to make the comparison fair?
3. How reliable is the LLM judge for low-resource languages like Old Uyghur or Yukaghir? The 100-sample human validation may not cover these cases adequately.

**Limitations:**

Yes

**Strengths And Weaknesses:**

Soundness
The benchmark construction is methodologically credible. It's great that they used professional historians to define difficulty levels rather than model performance, which meaningfully separates HistBench from prior work. The three-stage review pipeline and the dual LLM-plus-human evaluation metric add rigor. Both HistAgent and ODR-smolagents use GPT-4o, isolating architecture as the variable.
However, there are a few concerns. The GAIA generalization claim is not well supported. HistAgent uses Claude-3.7-sonnet while ODR-smolagents uses o1, making the comparison uncontrolled. The performance gap there could reflect base model differences rather than architectural advantages. This is a concrete flaw in an otherwise clean experimental design. Additionally, LLM judge reliability is only human-validated on 100 of 414 questions, with no breakdown by language or difficulty level. Given that the benchmark includes low-resource languages like Old Uyghur and Yukaghir, the judge's reliability on those cases is genuinely uncertain and could quietly affect reported accuracy numbers.

Presentation
The paper is well structured and easy to follow. The motivation is clearly articulated, the benchmark design is explained with appropriate detail, and the appendices are thorough. Figures 2 and 3 effectively communicate dataset diversity and difficulty definitions. The limitations section is honest.  A small note: the paper has the ICML submission formatting instructions on the header of each page, which should be removed.

Significance
This is where the paper is strongest. History has been almost entirely ignored in the LLM benchmarking literature despite being genuinely demanding. HistBench fills a real gap and does so with enough rigor that it could become a standard evaluation resource for the field. HistAgent demonstrates that targeted tool design can compensate for model size and match much stronger closed-source models like o3 at a fraction of the cost: a practically useful finding for researchers working in cost-sensitive settings. The broader implication, that the humanities deserve the same domain-specific agent treatment as chemistry, medicine, and law, is a valuable argument that could influence future research directions.

Originality
The paper borrows from smolagents and similar frameworks. The originality lies in the application of this architecture to history with carefully chosen domain-specific tools, and in the benchmark itself. HistBench is the first benchmark to embed historian judgment into difficulty classification, cover 29 languages and 5 modalities, and target historical reasoning as a distinct capability. That combination is genuinely new.

---

> ### Author Rebuttal · Authors · 2026-03-31
>
> > **Reviewer's comment**: Why does performance ...
>
> **Response**: We thank the reviewer for this insightful observation. The difficulty levels in HistBench are defined from the perspective of human historians, based on a structured rubric (e.g., multimodality, linguistic complexity, interdisciplinary scope), rather than model performance. As a result, difficulty is not expected to be monotonic for current LLMs or agents.
>
> Level 2 questions frequently involve fragmented cross-lingual reasoning that falls outside LLM pretraining coverage. For example, one task in Level 2 presents an English translation of a classical Chinese poem (from the 1975 anthology Sunflower Splendor) and asks for the original author (Wang Anshi); a Chinese reader familiar with classical poetry can readily recognize it from stylistic intuition, but all evaluated LLMs failed, as the English translation strips away the original linguistic cues and requires familiarity with a niche translation anthology rarely covered in pretraining data. In contrast, some Level 3 questions, though more complex by historian standards, anchor to high-frequency pretraining knowledge. Another task in Level 3 asks models to identify "samtso ačari" in an Old Uighur text—challenging for humans unfamiliar with Central Asian Buddhist philology—yet all models correctly answered Xuanzang, because "samtso" is a well-known phonetic rendering of 三藏 (Sanzang) and the surrounding context (Śīlabhadra, Nālandā) provides widely-documented corroborating cues. The difficulty LLMs experience thus reflects pretraining coverage as much as human-defined complexity—a distinction we will make explicit in the revision.
>
> > **Reviewer's comment**: The GAIA comparison uses different ... //
> > The GAIA generalization claim is not well supported.
>
> **Response**: We thank the reviewer for the insightful suggestion.
>
> We note that we already include **apple-to-apple comparisons on HLE**, where both HistAgent and ODR-smolagents use the same backbone, isolating the effect of agent design.
>
> For GAIA, HistAgent (Claude-3.7-sonnet) achieves **60.00% pass@1**, outperforming ODR-smolagents with o1 (55.15%). Notably, o1 is generally considered stronger than Claude-3.7-sonnet, indicating that HistAgent achieves better performance **with a weaker underlying model**, highlighting the effectiveness of the agent design.
>
> We further conduct a new matched-model experiment by running ODR-smolagents with Claude-3.7-sonnet, which yields **53.3% pass@1**, still below HistAgent (60.00%). We note that GAIA evaluation with ODR agent pipelines is computationally expensive (≈\$2.5 per question) and more expensive for HistAgent so it's hard to run full configuration like o1.
>
> | Agent            | Model                  | Average | Level 1 | Level 2 | Level 3 |
> |------------------|------------------------|---------|---------|---------|---------|
> | HistAgent        | Claude-3.7-sonnet      | **60.00** | 69.81  | 61.63  | 34.62  |
> | ODR-smolagents   | o1                     | 55.15   | 67.92  | 53.49  | 34.62  |
> | ODR-smolagents   | Claude-3.7-sonnet    | 53.3    | 60.38   | 55.81   |  30.77  |
>
> Thanks again for the insightful feedback, and we will include these new results in the revision.
>
> > **Reviewer's comment**: How reliable is the LLM judge... //
> > Additionally, LLM judge reliability is only
>
> **Response**: We thank the reviewer for this important concern. LLM-as-a-judge is a widely adopted paradigm for evaluating open-ended, multilingual tasks, including recent benchmarks such as CCFQA (Du et al., 2026), BenchMax (Huang et al., 2025), and HLE (Phan et al., 2025).
> For the specific cases raised (e.g., Old Uyghur, Yukaghir), we verify that they do not introduce ambiguity in evaluation: one is a multiple-choice question (deterministic evaluation), and the other is an **exact-match question** with both the original-language answer and an **English reference provided**. More broadly, most low-resource questions follow similar designs (multiple-choice or multilingual exact-match), reducing reliance on the judge’s language fluency.
>
> To assess reliability, we analyze our human validation set (100 questions), which covers diverse settings: 29/40/31 questions across Level 1/2/3, 41 multimodal questions, and ~4 cases involving low-resource or historical languages (e.g., Old Uyghur, Khitan script) out of 18 questions in the whole benchmark. This indicates that the validation is not limited to high-resource or simple cases.
>
> > **Reviewer's comment**: A small note...
>
> **Response**: We thank the reviewer for pointing this out. We will remove the ICML submission formatting instructions in the camera-ready version if the paper is accepted.

---

> > ### Author Rebuttal · Reviewer_7eNk · 2026-04-03
> >
> > My concerns have been adequately addressed.

---

### Official Review · Reviewer_aiZC · 2026-03-12

**Soundness:** 1
**Presentation:** 1
**Significance:** 1
**Originality:** 1
**Overall Recommendation:** 1
**Confidence:** 4

**Summary:**

Please see strengths and weaknesses

**Compliance With Llm Reviewing Policy:**

Affirmed.

**Key Questions For Authors:**

N/A

**Limitations:**

This paper does not have an impact statement.

**Strengths And Weaknesses:**

Per ICML author guidelines, this paper should be desk rejected for not including an impact statement. The authors left the template content for the Accessibility and Impact Statement in their paper. This is especially problematic since this paper covers a sensitive domain (history) and proposes an agent that might have significant impact on humanities scholarship.

---

> ### Author Rebuttal · Authors · 2026-03-31
>
> **Response**:
> We apologize for this formatting oversight. The broader impact discussion is already included in **Appendix E/F** of the submission, but is not placed in the required position. In the revised version, we will move it to the end of the main paper, before the references, to fully comply with ICML formatting requirements.
> We also note that, according to prior ICML FAQ guidance, missing or misplaced impact statements are not grounds for desk rejection. In our case, the content was already provided, and we will ensure it is correctly positioned in the revision.

---

> > ### Author Rebuttal · Reviewer_aiZC · 2026-04-06
> >
> > Thank you for your rebuttal. I appreciate your clarification that missing impact statements are not grounds for desk rejection alone. However, the fact that 1/3 page of placeholder text is left in the submitted manuscript suggests a lack of professionalism that should be disqualifying for the paper at a premium venue like ICML. It is hard to believe that the authors would not notice this if they did any proofreading, and even harder to believe that they took care to consider the ethical implications of their work (not to mention that Appendix F is insufficient and very generic). This is not respectful for the reviewers and the AC's time, and even more alarming given the sensitive nature of the paper's focus and its potential social impact as one of the early papers in this domain.

---

### Decision · Program_Chairs · 2026-04-30

**Decision:**

Accept (regular)

**Comment:**

HistBench: a benchmark of 414 expert-curated questions for multimodal historical reasoning. 29 languages (only 9 languages with 10+ questions). HistAgent: a specialized agent (tools: OCR, translation, and archival search) to tackle this along with empirical validation on existing models.

Rating: Reviewer aiZC (1*), Reviewer 7eNk (5), Reviewer mCfs (3), Reviewer 6Nsc (4). The reviewers like the motivation behind this benchmark and rigorous benchmark design resulting high diversity. The reviewers also like the proposed approach, finding the specialized agent effective, along with extensive experimental validation that comes with it.

*Note: Reviewer aiZC submitted their review late via email; their score has been updated to 2. Because the authors did not see this review during the rebuttal period, the AC treats it as additional unverified data point that the authors did not get to address.
Concerns

1. [Reviewer aiZC, Reviewer mCfs, Reviewer 6Nsc, partially resolved] Dataset Size and Language Long-Tailed Distribution. Some reviewers find 414 to be limited. The authors argue this is in line with benchmarks in the literature and that they aggressively filtered the candidates for quality rigor. Reviewer aiZC specifically highlights that 12 of the 29 languages have only 1 question (see Table 8 in Appendix B3). Top languages are English 228, Chinese 52, Classical Chinese 47, Russian 22, Japanese 13, French 10, Latin 8, German 8, which makes the distribution unbalanced and the claim “29 languages” technically correct but practically useless for low-frequency languages.

2. [Reviewer aiZC, Reviewer 7eNk, Reviewer mCfs, partially resolved] Unfair comparison and misleading reporting. The authors responded with new experimental results in some cases and claimed that they focus on same-backbone comparison (GPT-4o) to save cost and mainly analyze the effect of the agentic add-ons. The major cases are:

2.1 Model Mismatch: HistAgent uses Claude-3.7-sonnet on the GAIA validation set while baselines use o1.  The authors responded during the rebuttal by providing a new matched-model experiment to demonstrate that HistAgent still holds an advantage when the underlying model is controlled.

2.2 Pass@2: In Table 3, Reviewer aiZC notes the paper unfairly compares HistAgent's pass@2 against the baselines' pass@1 (best and second-best highlighted in each column). The authors did not see this comment.

3. [Reviewer 7eNk, Reviewer 6Nsc, somewhat resolved] Reliance on LLM Judges for open-ended multilingual historical reasoning, especially for low-resource languages. Reviewers questioned using LLMs for open-ended multilingual historical reasoning. The authors analyzed a 100-example validation subset and noted that exact-match or multiple-choice formats with English references make the evaluation deterministic for many low-resource cases. The authors also provided the basic coverage statistics for these 100 examples.

4. [Reviewer 7eNk, resolved] Reliability of Difficulty Level Performance. Model performance drops at Level 2 and rises at Level 3. The authors successfully argued that the difficulty level is based on human historians, which scales differently for AI (e.g., Level 2 requires fragmented cross-lingual reasoning outside standard pretraining).

5. [Reviewer aiZC, Reviewer 7eNk, Reviewer mCfs, Reviewer 6Nsc, mostly resolved] Ablations and Analysis. Reviewers wanted to see the exact contributions of individual, highly specific tools. The authors provided a partial ablation on a 30-question subset and a text-only vs. multimodal breakdown during the rebuttal.

6. [Reviewer aiZC, unseen and unaddressed] Real-world Utility and Ethics. aiZC noted a lack of discussion on how actual historians would use the tool (calling it a "technology looking for a problem") and criticized the cursory ethical considerations and placeholder impact statement.

Most of the concerns can be fixed with better presentation. However, some of them are either unseen or only partially addressed, including major ones regarding Pass@2, claims on language coverage, and the reliability of the judges. Thus, the AC recommends weak acceptance. The authors should take into consideration these feedbacks: 1) tone down some of their claims, 2) provide clearer experimental comparison with fair interpretation, 3) include evidence to show the judges' reliability on low-resource languages, 4) expand the discussion on real-world utility and ethics.